# Cloud Condensation Nuclei properties of South Asian outflow over the northern Indian Ocean during winter

Vijayakumar S Nair[1], Jayachandran Venugopalan Nair[1], Sobhan Kumar Kompalli[1], Mukunda M Gogoi[1] and S Suresh Babu[1]

[1]Space Physics Laboratory, Vikram Sarabhai Space Centre, Thiruvananthapuram, India
*Correspondence to*: Vijayakumar S Nair (vijayakumarsnair@gmail.com)

**Abstract.** Extensive measurements of cloud condensation nuclei (CCN) and condensation nuclei (CN) concentrations in the South Asian outflow to the northern Indian Ocean were carried out on board an instrumented research vessel, as part of the Integrated Campaign for Aerosols, gases and Radiation Budget (ICARB) during winter season (January-February 2018). Measurements include a north-south transect across the South Asian plume over the northern Indian Ocean and east-west transect over the equatorial Indian Ocean (~2 °S), which is far away from the continental sources. South Asian outflow over the northern Indian Ocean is characterized by the high values of CCN number concentration (~5000 $cm^{-3}$), low CCN activation efficiency (~25%) and steep increase in CCN concentration with an increase in supersaturation. In contrast, low CCN concentration (~1000 $cm^{-3}$) with flat supersaturation spectra was found over the equatorial Indian Ocean. The CCN properties exhibited significant dependence on the geometric mean diameter (GMD) of the aerosol number size distribution and CCN activation efficiency decreased to low values (< 20%) at the times of new particle formation events over near-coastal and remote oceanic regions. The analysis of the activation fractions for the 'similar' aerosol size distributions over the northern Indian Ocean indicated the primary role of aerosol number size distribution, followed by the chemical composition on CCN activation efficiency. The dependence of CCN properties and activation fraction on size segregated aerosol number concentration, especially during the ultrafine particle events, is investigated in detail for the first time over the region.

## 1 Introduction

Aerosol-climate interaction is one of the major uncertain components of the Earth-atmosphere system, which includes several pathways like aerosol-radiation (scattering and absorption of solar radiation), aerosol-cloud (modification of cloud properties due to aerosols), aerosol-cryosphere (snow albedo reduction due to aerosol deposition) and aerosol-biosphere interactions having significant radiative forcing (IPCC, 2013; Rosenfeld et al., 2014; Li et al., 2016). IPCC (2013) has estimated the global mean radiative forcing due to aerosol-radiation and aerosol-cloud interaction as -0.9 (-1.9 to -0.1) W $m^{-2}$, which compensate nearly 30% of the warming due to well-mixed greenhouse gases. The fundamental parameter

relevant for understanding the aerosol-cloud interaction is the cloud condensation nuclei (CCN), which are those aerosols that get activated at supersaturations pertinent to atmospheric conditions (Rosenfeld et al., 2014). Hence, the large uncertainty in the estimates of aerosol-cloud interaction points to the necessity of dedicated field campaigns and modelling efforts to improve the level of scientific understanding on CCN activation and to accurately quantify the change in microphysical properties of clouds due to anthropogenic aerosols (Rosenfeld et al., 2014). Regional meteorology also plays a major role in the aerosol-cloud interaction and the effect of aerosols on clouds varies with meteorological regimes (Reutter et al., 2009; Kerminen et al., 2012; Schmale et al., 2018). Since, the oceans cover about 70% of the Earth surface and the low-level marine clouds (stratus and stratocumulus) are highly sensitive to aerosol perturbations (Rosenfeld et al., 2014), understanding the CCN properties and its dependence on physicochemical properties of aerosols is crucial in understanding the aerosol-cloud interactions over the regions lying in the downwind of continental outflow (Furutani et al., 2008, Kim et al., 2014, Snider and Brenguier, 2000).

South Asian region, especially northern India, experiences high aerosol loading during winter season (Nair et al., 2007; Bharali et al., 2019). These aerosols are mostly confined in the atmospheric boundary layer and have significant implications on the air quality, visibility, human health and radiation budget (Lelieveld et al., 2001; Bharali et al., 2019). Due to the favourable prevailing wind system, these continental aerosols are being transported over the northern Indian Ocean (Arabian Sea and Bay of Bengal), and the effects of these anthropogenic aerosols on regional climate has been the major scientific theme for the several field campaigns and modelling studies carried out during the last two decades (Moorthy et al., 2009; Ramanathan et al., 2001; Lelieveld et al., 2001; Ackerman et al., 2000, Nair et al., 2013). South Asian outflow characterized during the Indian Ocean Experiment (INDOEX) revealed high concentrations of black carbon (BC) and organic carbon (OC) aerosols emitted from biomass and fossil fuel burning over South Asia during winter. Several studies have shown that the transport of anthropogenic aerosols, especially carbonaceous aerosols, to the marine atmosphere perturbs the regional radiation balance through aerosol-radiation (Ramanathan et al., 2001; Moorthy et al., 2009) and aerosol-cloud interactions (Ackerman et al., 2000, Chylek et al., 2006). The INDOEX observations indicated that CCN characteristics of the northern Indian Ocean are impacted by the outflow of South Asian aerosols as seen from the widespread nature of the high CCN number concentrations over the region (Cantrell et al., 2000; 2001). Following the INDOEX, even though there were several field experiments like ICARB-2006, ARMEX (Arabian Sea Monsoon Experiment) and ICARB-2009 to characterize the aerosol properties, simultaneous measurements of the CCN and aerosol properties were never attempted in the South Asian outflow region for the last two decades.

Even though, several studies have addressed the activation properties of aerosols using extensive measurements over distinct aerosol environments (Furutani et al., 2008; Rose et al., 2011; Kerminen et al., 2012; Pöhlker et al., 2016; Schmale et al., 2018), studies are limited over South Asia and its outflow regions. The CCN properties also exhibit large spatial and temporal heterogeneities similar to that of extrinsic properties of aerosols (Andreae, 2009; Jefferson, 2010). Hence, aerosol-CCN studies at distinct aerosol types (like polluted, marine, biogenic and dust) assume importance (Schmale et al., 2018). Size segregated measurements of CCN concentration along with physical and chemical properties of aerosols

are essential to unravel the complex dependence of size and chemistry of aerosols on CCN activation (Dusek et al., 2006a; Pöhlker et al., 2016; Rose et al., 2011; Furutani et al., 2008), which is essential for the better prediction of CCN concentration, and thus to reduce the climate forcing uncertainty due to aerosol-cloud interactions. Studies on relative importance of aerosol size distribution and chemical composition (mixing state) on the CCN properties are rather limited over the Indian sub-continent, except few studies (Jayachandran et al., 2018 references are therein). At microscopic level, a fraction of the total aerosols get activated as CCN which depends primarily on the size, followed by chemical composition and mixing state of the aerosol system (Jimenez et al. 2009; Dusek et al., 2006a; Kerminen et al., 2012; Rose et al., 2011). There exists a large uncertainty in the activation properties of particles, especially the contribution of ultrafine particles (size below 100 nm) and new particle formation events to the global CCN concentration (Pierce and Adams, 2007; Merikanto et al., 2009). Further, the role of higher levels of ultrafine aerosols on CCN number concentration and activation fraction is not investigated over the northern Indian Ocean, especially when the continental outflow dominates the marine aerosol system.

In this study, we present the results from the dedicated shipborne measurements onboard the oceanographic research vessel (ORV) Sagar Kanya (SK) during winter 2018 (hereafter ICARB-2018) carried out as part of Integrated Campaign for Aerosols gases and Radiation Budget (ICARB) experiment with a broad objective to characterize the South Asian outflow. The present study focusses on the wintertime measurements of the CCN concentrations at different supersaturations, along with simultaneous measurements of the aerosol properties when the entire northern Indian Ocean was under the influence of continental outflow from South Asia. The measurements were carried out within the continental outflow and remote oceanic regions far away from South Asia. The northern Indian Ocean is an ideal and unique region to study the role of anthropogenic aerosols on CCN activation, where aged continental plume having high concentration of carbonaceous aerosols and volatile vapours mix with marine aerosols during the Northern hemispheric winter (Mayol-Bracero et al., 2002; Nair et al., 2007; 2013). This paper discusses the general characteristics of the CCN concentrations, activation properties, and their association with aerosol number size distribution at different regions of the South Asian outflow. The relative role of aerosol number size distribution on the variability of CCN activation and the contribution of ultrafine particles to the CCN concentration will be discussed in detail.

## 2 Campaign, instruments and general meteorology

During the winter months, the prevailing large-scale circulation over South Asia is favourable for the transport of anthropogenic aerosols to the Arabian Sea and Bay of Bengal (Ramanathan et al., 2001; Nair et al., 2007; Moorthy et al., 2009). The spatial extend of the aerosol transport to the Indian Ocean is qualitatively depicted by the climatological (2002-2017) mean aerosol optical depth (AOD) derived from MODIS observations over the northern Indian Ocean (contours in Figure 1). High aerosol loading (AOD > 0.3) is observed over the northern Bay of Bengal and southeastern Arabian Sea during winter. The ICARB-2018 measurements were planned to carry out within this plume area over Southeastern Arabian Sea and the regions far away from the continental outflow. The ship cruise on-board ORV Sagar Kanya started from Goa (15°N, 73.8°E) on 16 January 2018 and ended at Tuticorin (8.8°N, 78.2°E) on 13 February 2018. The cruise track is shown

in Figure 1. ICARB-2018 cruise experiment has mainly three phases: (*i*) latitudinal (16-22 January 2018), (*ii*) longitudinal (23-31 January 2019), and (*iii*) return (01-13 February 2019) phases. Aerosol properties reported during each of the measurement phases has strong association with large-scale meteorology, as reported in the earlier field experiments (Lelieveld et al., 2001; Ramanathan et al., 2001; Nair et al., 2013). Depending on the origin of airmasses, measurements made during ICARB-2018 are classified into three different groups. The first phase of the cruise over southeastern Arabian Sea (SEAS) is divided into 'SEAS1' and 'SEAS2' regions where the former is influenced by the airmasses from peninsular India, and the latter is from the Bay of Bengal. The second phase of the cruise over the remote equatorial Indian Ocean (EIO), where the continental influence is rather less compared to SEAS1 and SEAS2, is considered as the third group. Mostly calm and clear sky conditions prevailed during the campaign except for few rainfall spells during 4, 6 and 7 February 2018 when the ship was sailing over the southeastern Arabian Sea (4° N, 67.2° E). These wide spread rainfall events associated with the western disturbances are also observed over the peninsular and western part of the Indian sub-continent during this period due to the. Due to these weather events, we have not considered the data collected during the return phase (phase 3) in this study. The mean values of air temperature and relative humidity during ICARB-2018 from the automatic weather station observations were 28.0 ± 0.8°C and 74 ± 5% respectively.

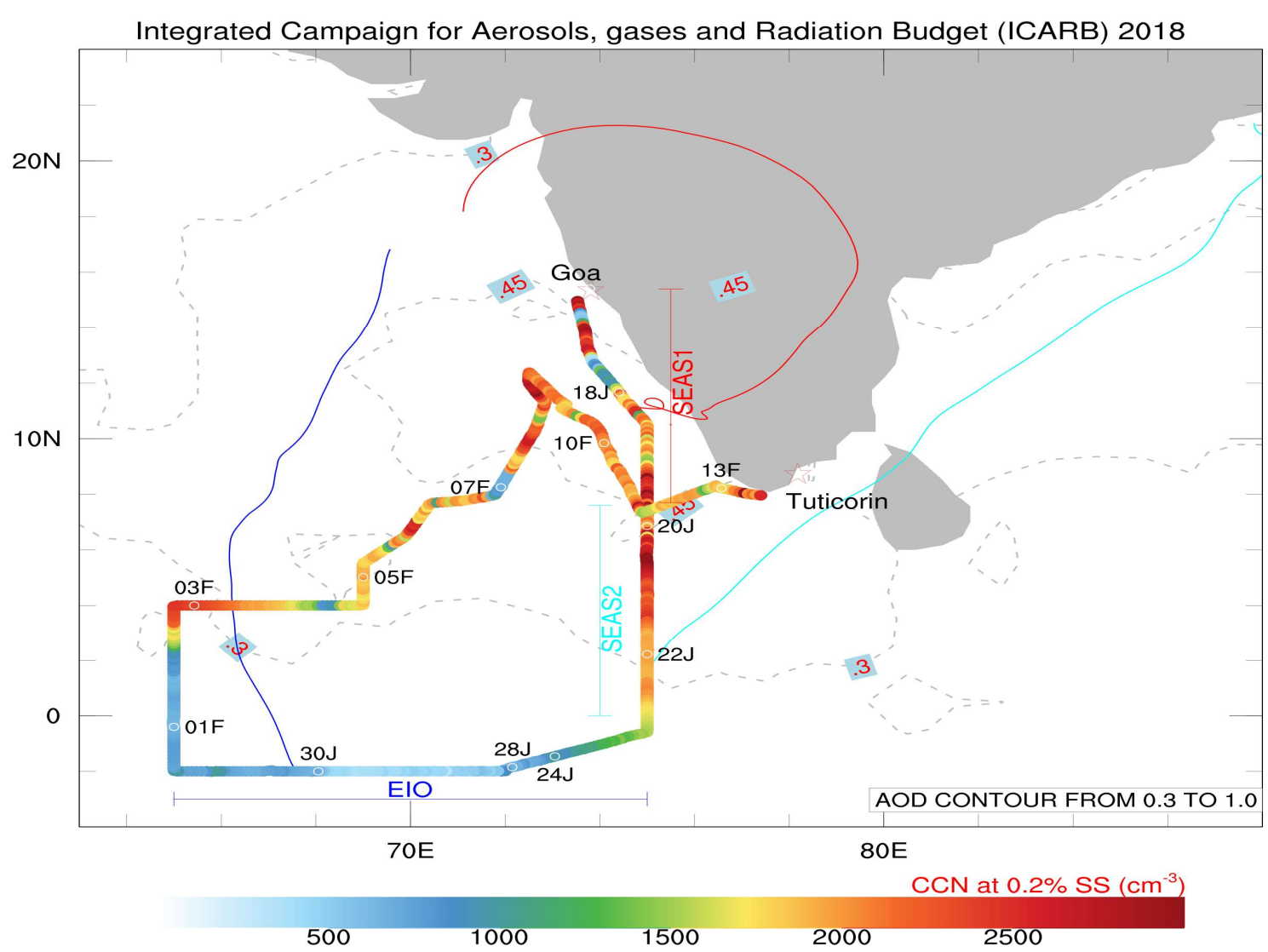

**Figure 1:** The cruise track of ICARB 2018 over northern Indian Ocean. The colour of the cruise track indicates the CCN number concentration ($cm^{-3}$) at 0.2% supersaturation measured onboard the ship during ICARB-2018. CCN characteristics at three distinct regions marked as SEAS1, SEAS2 and EIO are discussed in this study. Noontime ship location is marked by dates suffixed with J and F, where 'J' for January and 'F' for February. Grey coloured contours indicate the climatological mean values of aerosol optical depth at 550 nm derived from MODIS satellite for winter season. Typical airmass back trajectories arriving at the ship location on 18, 22, and 30 January, representing the SEAS1 (red color), SEAS2 (cyan color) and EIO (blue color) region respectively, estimated using HYSPLIT model are shown as solid lines.

Aerosol instruments were installed in a customized laboratory on the top deck (~15 m above sea level) of the ship. The instruments aspirated ambient air from a manifold sampling inlet having 10 µm size cut off at a flow rate of 16.6 litres per minute. The flow rate was frequently monitored and maintained using a flow controller and external pump. In order to avoid the contamination of the ship's exhaust on the aerosol measurements, the bow of the ship was mostly aligned to the upwind direction, and those measurements when the wind was blown from the rear side of the ship were excluded. All the measurements were averaged in hourly basis for uniformity after applying the necessary quality checks, instrument specific data correction procedures, and then geo-located using the time stamped position information available from the GPS receiver installed onboard. The simultaneous measurements of CCN number concentration using CCN-counter and aerosol number size distribution from scanning mobility particle sizer (SMPS) form the basic dataset for this study.

CCN number concentration measurements at different supersaturations ranging from 0.2 to 1.0 % were carried out using a single column continuous flow CCN counter (Model: CCN-100, Make: DMT) at a time resolution of 1 Hz. The difference in the radial diffusion rate of heat and water vapour is made use to develop specific supersaturations along the centre line of the instrument column, depending on the sheath to sample flow rate and the temperature gradient along the column (Roberts and Nenes, 2005). The aerosols are introduced into the centreline of the filtered, humidified sheath flow. The aerosols of size greater than the critical diameter at the set supersaturation will grow as CCN, which are counted further by an optical particle counter having diode laser source at 660 nm wavelength. In the present study, a constant flow rate of 0.5 LPM and a steady sheath to sample flow ratio of 10:1 was maintained throughout the campaign. Duration of each cycle of CCN measurements (0.2 to 1.0% supersaturation) spanned for 30 minutes, giving more time for the lowest supersaturation (0.2%) (Jayachandran et al., 2017, 2018). Considering the instability inside the column during the supersaturation changes, first 2 minutes data of each supersaturation are excluded from the analysis.

Aerosol number size distribution (NSD) measurements were carried out using a scanning mobility particle sizer (SMPS, Make: TSI), which measure the size segregated number concentration of particles from ~9 nm to 420 nm. The SMPS consist of a differential mobility analyser (DMA, TSI 3081) and a water based condensation particle counter (CPC, TSI 3786). The particles segregated according to their electrical mobility by the DMA are allowed to grow in the condensation chamber of CPC to optically detectable size range. These particles are counted by using an optical particle counter (Wang and Flagan, 1990). All the instruments were calibrated prior to the campaign following the standard protocols.

## 3 Results and discussion

### 3.1 CCN number concentrations

The number concentration of CCN at 0.2% supersaturation, which represent the concentration of aerosols that are hygroscopic and sufficiently large enough (size >100 nm) to get activated at low supersaturations, is shown along the cruise

track in Figure 1. High concentrations of CCN (~ 2000 cm$^{-3}$) are observed in the plume area (SEAS1) and lower CCN values (<500 cm$^{-3}$) are seen over the southwest part of the cruise track (~2°S, 65 to 72°E). The low values observed during the return phase of the cruise are attributed to the effective wet scavenging of aerosols due to thunder storms experienced on 4, 6 and 7 February 2018. The SEAS2 region which is mostly affected by the advection from the Bay of Bengal region (lying in the downwind of the Indo Gangetic Plain, Figure 1), also showed high CCN concentrations up to 2500 cm$^{-3}$ indicating the widespread influence of South Asian outflow. The latitudinal gradient of CCN at different supersaturations from 15 °N to the equator, between the longitudes 74 °E to 75 °E, shows distinctly different patterns for the CCN concentrations at low (0.2 %) and high (1.0 %) supersaturations as shown in Figure 2. At 0.2 % supersaturation, CCN values observed over SEAS1 (1683 ± 435 cm$^{-3}$) are lower or comparable to that over SEAS2 (1868 ± 276 cm$^{-3}$). This is in contrary to the CCN values at 1.0 % supersaturation, where CCN values are two-fold higher over SEAS1 (5954 ± 1493 cm$^{-3}$) compared to SEAS2 (2513 ± 668 cm$^{-3}$). The systematic decrease in the CCN number concentration with latitude is clearly seen below the southern tip (< 8 °N) of the peninsular India, whereas, measurements above 8 °N are modulated by the heterogeneous sources located along the west coast of India. Over SEAS1, the CCN concentrations increased by almost 5 to 7 folds when supersaturation changed from 0.2 % to 1.0 %, whereas CCN increase with supersaturation is insignificant over SEAS2 and the equatorial Indian Ocean.

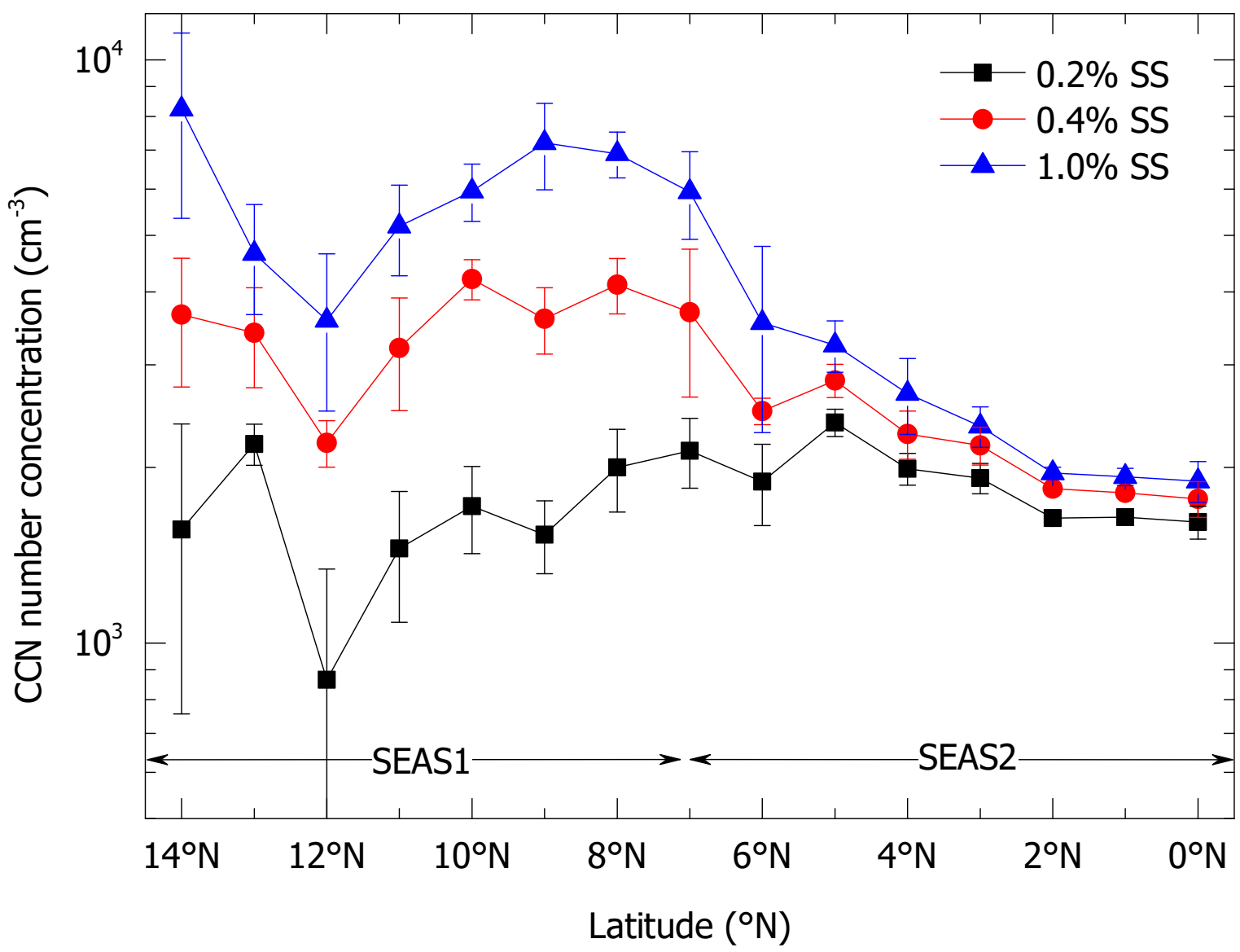

**Figure 2:** Latitudinal variation of the CCN number concentration at 0.2%, 0.4% and 1.0% supersaturations over southeastern Arabian Sea (SEAS). The vertical bars on each data point indicate the standard deviation of the measurements. The SEAS region is further classified into SEAS1 and SEAS2.

The regional variation of the mean CCN concentration with respect to supersaturation (normally called as CCN spectra) over SEAS1, SEAS2 and EIO are shown in Figure 3. The mean CCN spectra observed during December 2017 at Thumba (8.5°N, 77°E, Jayachandran et al. 2018), which is located in the west coast of peninsular India (close to SEAS1) and experienced the same synoptic conditions as that during the ICARB-2018 campaign, is also shown in the figure for comparison. The variation of CCN with supersaturation is parameterised using the Twomey's empirical relation, $N_{CCN} = C\ S^k$ where S is supersaturation and k is Twomey's exponent which is also mentioned in the figure. Twomey's exponent (k) indicate the qualitative information on the CCN active aerosol size distribution, with high k values implying the dominance of ultrafine mode aerosols and low k values indicating the accumulation mode aerosol dominance in the measured NSD (Fang et al., 2016; Jayachnadran et al., 2017; Gunthe et al., 2009). As shown in Figure 2, the CCN concentration measured in the South Asian outflow is very sensitive to the supersaturation over SEAS1 (coastal regions adjoining India), whereas CCN concentration is less dependent on supersaturation over the regions far away from the continental sources (SEAS2 and EIO). The higher (0.83 ± 0.22) value of Twomey's exponent (k) are over SEAS1 (steep CCN spectra) is attributed to the dominance of ultrafine mode aerosols (or hydrophobic aerosols) in the continental outflow. The lower k values of 0.21 ± 0.19 and 0.13 ± 0.11 estimated for SEAS2 and EIO respectively suggested larger particle dominance due to extended transit of aerosols over the oceanic regions leading to enhanced sizes.

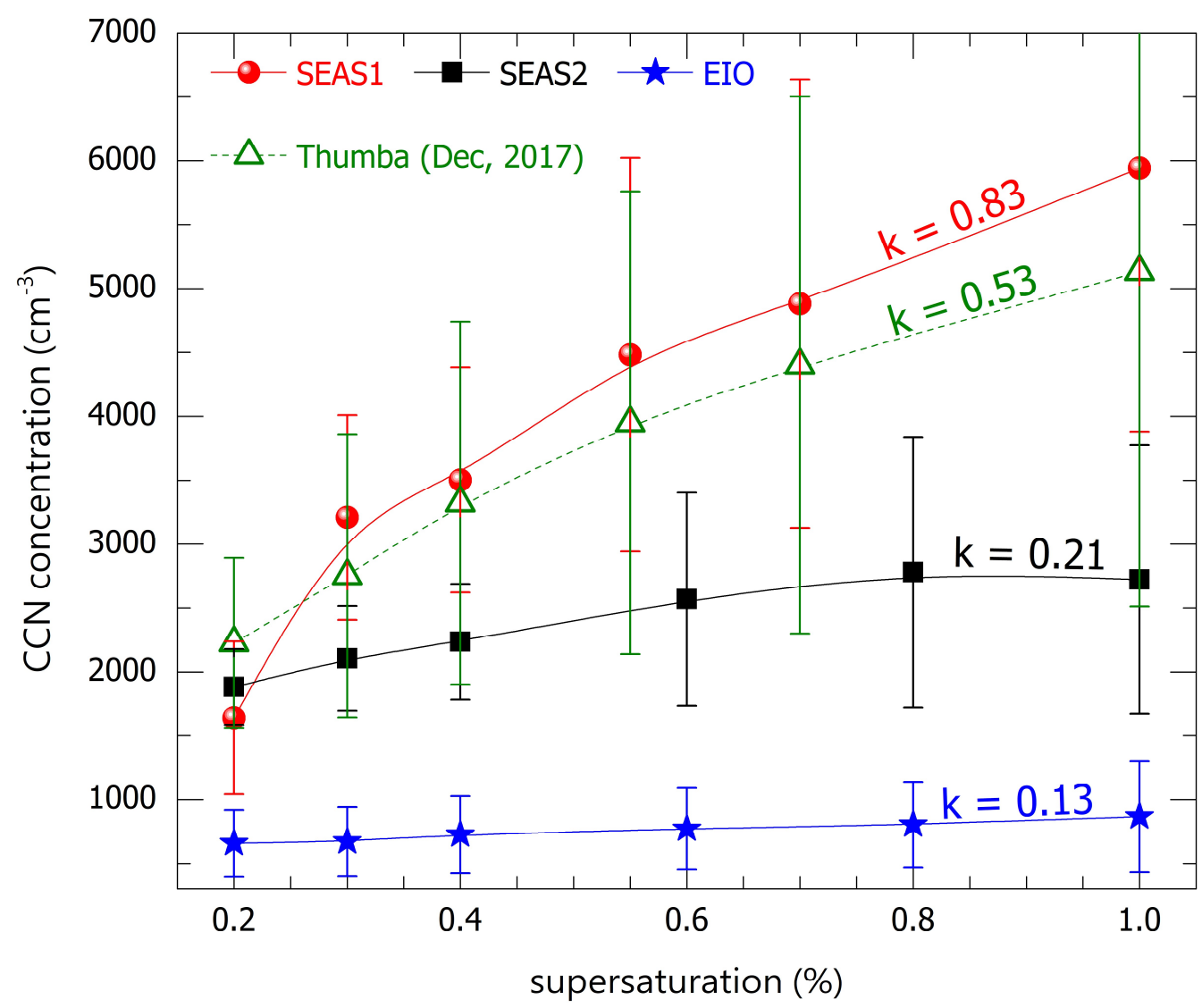

**Figure 3:** Mean CCN number concentration with the standard deviation at different supersaturations over northern Indian Ocean (SEAS1, SEAS2, and EIO) during ICARB-2018. The CCN spectra measured at a coastal station, Thumba are also shown in figure for comparison.

It would be interesting to compare the present values of CCN with the previous observations, which are rather scarcely reported over the oceanic regions surrounding South Asia. Based on the shipborne measurements during INDOEX, Cantrell et al., (2000) reported the CCN concentration of 1000 to 2500 $cm^{-3}$ (at 0.5 % supersaturation) over 10 °N to 15 °N. Similarly, airborne measurements on-board research flight (NCAR C-130) during INDOEX reported the values of CCN at 1.0% supersaturation as ~1000 $cm^{-3}$ for polluted airmasses and less than 100 $cm^{-3}$ for clean marine conditions over southern Indian Ocean (Hudson and Yum, 2002). The CCN values over SEAS1 region during ICARB-2018 are (i) higher than the CCN values reported during INDOEX, (ii) comparable to the values at the coastal site (Thumba) in southern peninsular India, and (iii) less than the values reported from continental polluted locations in the Indo Gangetic Plain (Jayachandran et al., 2017). The CCN values observed over SEAS are mostly lower or comparable to the values over polluted continental sites (2900 ± 2800 $cm^{-3}$ at 0.4 % supersaturation) as reported by Andreae (2009) based on the compilation of CCN observations carried out worldwide. Interestingly, the CCN values over EIO (which is relatively remote oceanic region) during ICARB-2018 are almost 7 fold higher than the clean marine CCN values (107 ± 56 $cm^{-3}$) and lower than the polluted marine conditions (1060 ± 400 $cm^{-3}$) reported by Andreae (2009). Cantrell et al., (2001) also reported high CCN concentrations (in the range of 300 to 1000 $cm^{-3}$) at 0.3 % supersaturation over Kaashidhoo climate observatory over the equatorial Indian Ocean during February-March 1999. Higher concentration of CCN at the equator and further south of it clearly indicated the wide spread influence of continental outflow over this region (Ramanathan et al., 2001).

In general, as we move away from the continental sources, aerosol abundance (AOD, black carbon, and total mass concentration) decrease towards open ocean as reported by several studies over the northern Indian Ocean (Ramanathan et al., 2001; Moorthy et al., 2009). Similarly, Chylek et al., (2006) have reported a latitudinal decrease in the CCN concentration at 1% supersaturation from 1850 $cm^{-3}$ at 4 °N to 700 $cm^{-3}$ at 0 °N over the Indian Ocean during the INDOEX campaign. Over the same latitudinal sector, the rate of decrease of CCN concentration with latitude was lower during ICARB experiment (1.4 times) compared to the INDOEX values, which highlighted the persistent and widespread impact of continental outflow over the northern Indian Ocean during ICARB-2018. In addition, CCN variation with supersaturation also depicted a latitudinal gradient with high k values over SEAS1 and low values over EIO. It is interesting to note that, irrespective of the large decrease in aerosol loading and change in aerosol microphysical properties over the region, the CCN concentrations at 0.2% are comparable at all locations (SEAS1, SEAS2 and Thumba) except over EIO (Figure 3). The high k values are observed at the coastal site, Thumba (Jayachandran et al., 2018) and measurements from Kaashidhoo Climate Observatory (4.97° N, 73.5° E) during INDOEX (Cantrell et al. 2001). Similar to SEAS1, an average k value of 0.8 was reported over North-East Atlantic by Snider and Brenguier, (2000) during ACE2 campaign. Schmale et al., (2018) also reported similar finding based on the data collected from several distinct locations in Europe. Since the k value of the CCN spectrum depends highly on the dominance of the ultrafine particles in the aerosol size distribution and hygroscopicity of the aerosol system, the k values estimated from the CCN measurements for a short range of supersaturations should be interpreted carefully. Generally, primary carbonaceous particles and newly formed ultrafine particles have low contribution

to the CCN concentration and require extremely high supersaturation conditions for the CCN activation (Dusek et al., 2006b, Pierce and Adam, 2007).

### 3.2 CCN activation fraction and geometric mean diameter

The fraction of total aerosol concentration (CN) that can act as CCN at a specific supersaturation (CCN(S)/CN) is termed as CCN activation fraction (AF(S)) or activation efficiency, which is governed mainly by the number size distribution and composition of the aerosol system (Schmale et al., 2018; Dusek et al., 2006a). The scatter plot between the CN and CCN over SEAS1, SEAS2 and EIO is shown in Figure 4 (Top panel) and the colour scale indicate geometric mean diameter (GMD) corresponding to the composite aerosol number size distribution. CCN being a specific subset of CN, the CCN concentrations increased with increasing total aerosol concentrations during most of the observations. Regression analysis of CN and CCN at 0.4% (slope ~ 0.2 and $R^2$ ~ 0.45) and 1.0% (slope ~ 0.29 and $R^2$ ~ 0.44) supersaturations has poor association during the entire campaign period. This is in contrast to the earlier observations (Gunthe et al., 2009; Jayachandran et al., 2018), where association between CN and CCN has increased with supersaturation. The association between CCN and CN weakened when the CN concentration is very high and fine mode aerosols contributed significantly to the CN concentrations, as evident from the lower values of GMD of the aerosol size distribution (Figure 4a colour scale). When the GMD values are greater than 100 nm, CCN and CN followed an excellent relationship with $R^2$ ~ 0.99 and mean activation fraction of 69 ± 10% over all the regions (SEAS1, SEAS2 and EIO). This clearly highlighted that most of the particles in this size range are get activated irrespective of the regional heterogeneities in the aerosol composition. Aerosol system having GMD less than 60 nm (when the aerosol number size distribution is dominated by the ultrafine particles) is deviated significantly from regression line. This implies that the abundance of ultrafine particles has direct impact on activation fraction, since most of these particles may not get activated at 0.4% and 1.0% supersaturation levels (Pierce and Adam, 2007).

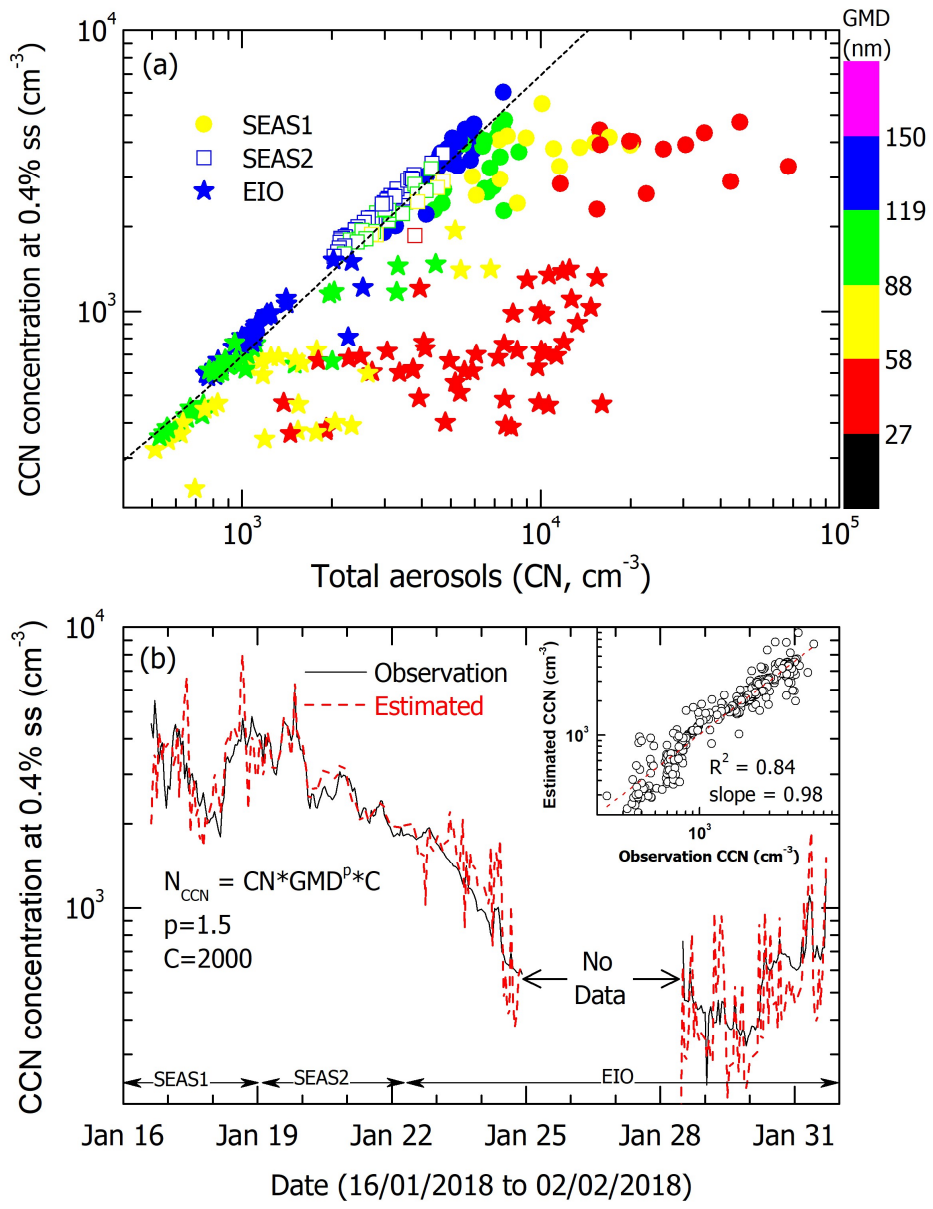

**Figure 4:** (a) Scatter plot between total aerosol (CN) and CCN at 0.4% supersaturation over SEAS1, SEAS2 and EIO. Colour shows the geometric mean diameter (GMD) of the aerosol size distribution. Regression fit for GMD greater than 100 nm is shown as black dotted line. (b) CCN concentration at 0.4% supersaturation measured using CCN counter and estimated using the empirical relationship between total aerosol number concentration and geometric mean diameter (GMD). The 'p' and 'C' are empirical constants estimated using the regression technique.

The influence of GMD on the CN-CCN association is further investigated using the regression analysis between the observed CCN at 0.4% supersaturation with the product of CN and GMD (CN*GMD). The correlation coefficient ($R^2$) has improved from 0.44 to 0.84 when CN multiplied with GMD. The CCN concentration is estimated from the empirical relationship between CN and GMD ($CCN_{est}$ = C*CN*GMD where C is a constant), which showed very good association with the measured CCN concentration (at 0.4% supersaturation). The regression coefficient between measured and estimated CCN further increased to 0.94 for a power law, $CCN_{est} = CN*GMD^{p}*C$ where p (1.5) and C (2000) are constants estimated iteratively for the highest value of $R^2$. The temporal variation and scatter plot of measured and estimated CCN concentrations at 0.4% supersaturation are shown in Figure 4b. By accounting the effect of GMD on CCN concentration, this analysis clearly demonstrated the primary role of aerosol number size distribution on CCN activation. Since the ultrafine mode aerosols have lesser hygroscopicity compared to the accumulation and coarse mode aerosols (Pohlker et al., 2016; Gunthe et al., 2009; Pierce and Adams, 2007; Rose et al., 2011), the empirically estimated CCN number concentration is overestimated during the periods of ultrafine particles dominance in the size distribution. The low activation fractions during the new particle formation events and for the higher abundance of ultrafine mode aerosols have been reported in literature (Pohlker et

al., 2016). The present analysis of estimating the CCN concentration from the product of CN and GMD for widely varying aerosol size distribution is analogues to the better association between the product of aerosol scattering coefficient and its spectral dependence (angstrom exponent) with CCN concentration rather than the regression between scattering coefficient and CCN concentration (Jayachandran et al., 2018).

In general, aerosol size distribution plays a major role in CCN activation. The ultrafine particles significantly decrease the activation fraction compared to the accumulation mode aerosols. Since the CN number concentration decreased from southeastern Arabian Sea to the equatorial Indian Ocean, delineating the periods of ultrafine particle dominance from a threshold CN value is difficult (see Figure 4a). The CN values over the equatorial Indian Ocean (> 5000 $cm^{-3}$) during ultrafine particle events are lower than the CN values even without ultrafine particle events over southeastern Arabian Sea region. In Figure 5, the association of CCN activation fraction with the CN number concentration is shown where the values are shown in different colours for different regions. For a constant CN concentration (5000 cm-3), the activation fraction of EIO (~15%) is much lower than that of SEAS1 (~65%) due to the presence of ultrafine particles. Figure 5 highlighted that the activation fraction decreases with CN following power law dependence ($AF = a*CN^b$, where a and b are constants) with similar coefficients ($b = -0.84$, $R^2 = 0.85$) over both SEAS1 and EIO regions. Yum et al., (2007) also showed that CCN efficiency decreases with increase in ultrafine particle concentrations irrespective of airmass type, which implies the lower hygroscopicity of these particles irrespective of the chemical composition (Pierce and Adams, 2007).

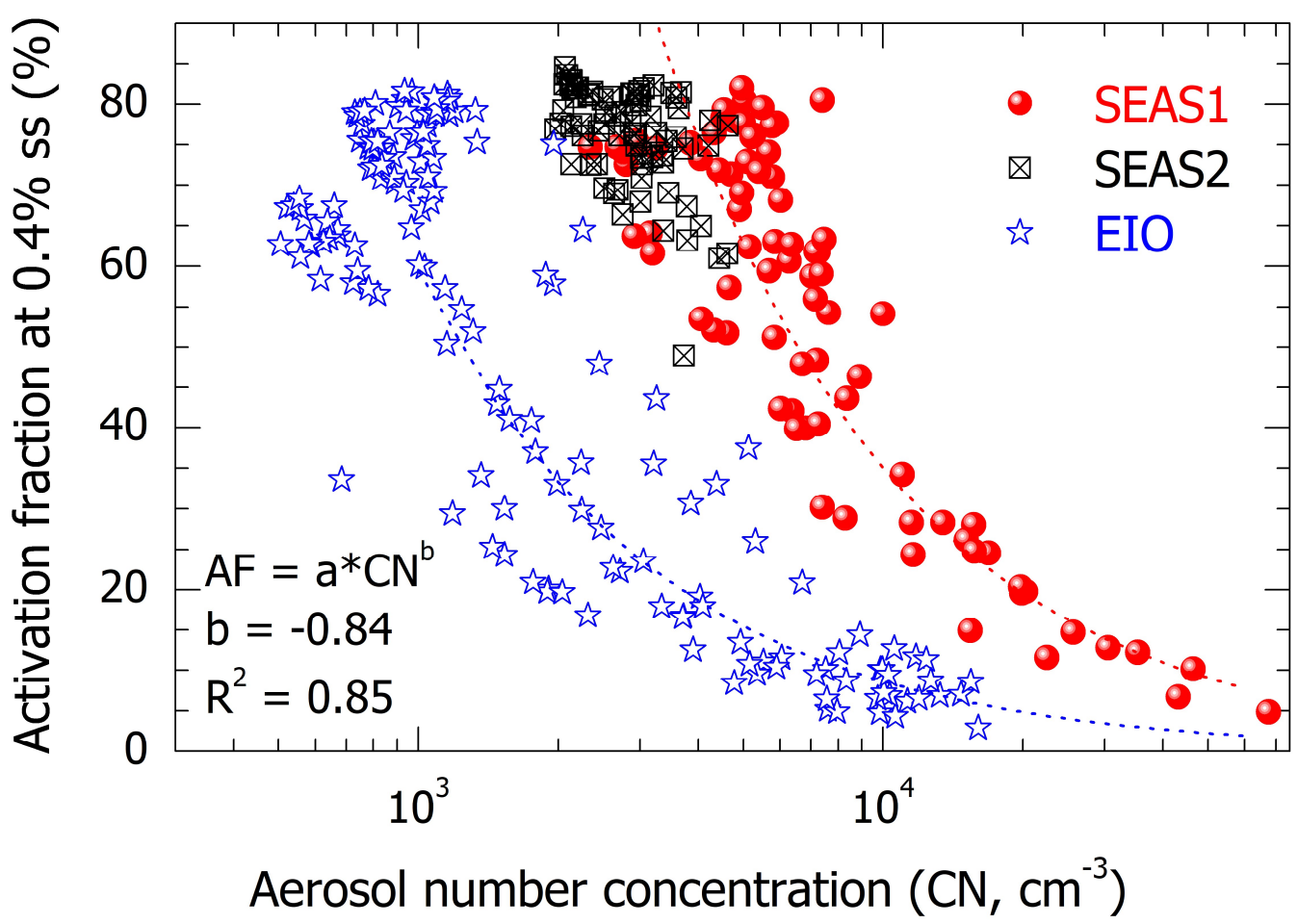

**Figure 5:** Association of activation fraction at 0.4% supersaturation with CN number concentration during SEAS1, SEAS2 and EIO of ICARB-2018. Dotted lines indicate regression fit to the data.

The scatter plot between the CCN activation efficiency at 0.4% supersaturation and the GMD of aerosol size distribution segregated for different marine regions is shown in the Figure 6a. It is clear that higher CCN activation efficiency is observed for particle size distributions dominated by larger particles (GMD > 130 nm) irrespective of the regions and airmass pattern. Wherever the contribution of ultrafine particles to the total aerosol concentration is significant,

the activation fraction decreased. In general, low (high) GMD values resulted in low (high) CCN activation fraction, irrespective of aerosol composition, in the south Asian outflow. For a given GMD of 90 nm, the activation fraction varied over a wide range from 20 to 60%, which is more notable over the EIO region (stars), where the ultrafine particles contribute 20% to 50% to the total number concentrations. Earlier, Kim et al., (2014) have also observed a significant spread in the association between CCN at 0.6% supersaturation and GMD in the range of 40-70 nm, which is attributed to the heterogeneities in aerosol chemical composition. The association between GMD and activation efficiency weakened during the new particle formation events and aerosol size distribution having multiple modes, especially in the coarse and ultrafine particle regime. Supersaturation spectra of CCN activation fraction for low (25-50 nm) and high (125-150 nm) values of GMD are shown in Figure 6b. In contrast to SEAS1 and EIO, low GMD cases were not observed over SEAS2. For low GMD cases, activation efficiency even at 1.0% supersaturation is mostly less than 30%, which implies the low hygroscopicity of the ultrafine particles. In contrast, higher activation efficiencies (> 60%) are observed for high GMD cases over all the regions irrespective of the supersaturation conditions. The very low activation efficiency at 0.2% supersaturation (~30%) observed over SEAS1 region increased drastically to ~100% at 1.0% supersaturation for high GMD conditions. Regionally, SEAS1 aerosols are more CCN active especially at higher supersaturations (1.0%) compared to the EIO aerosols.

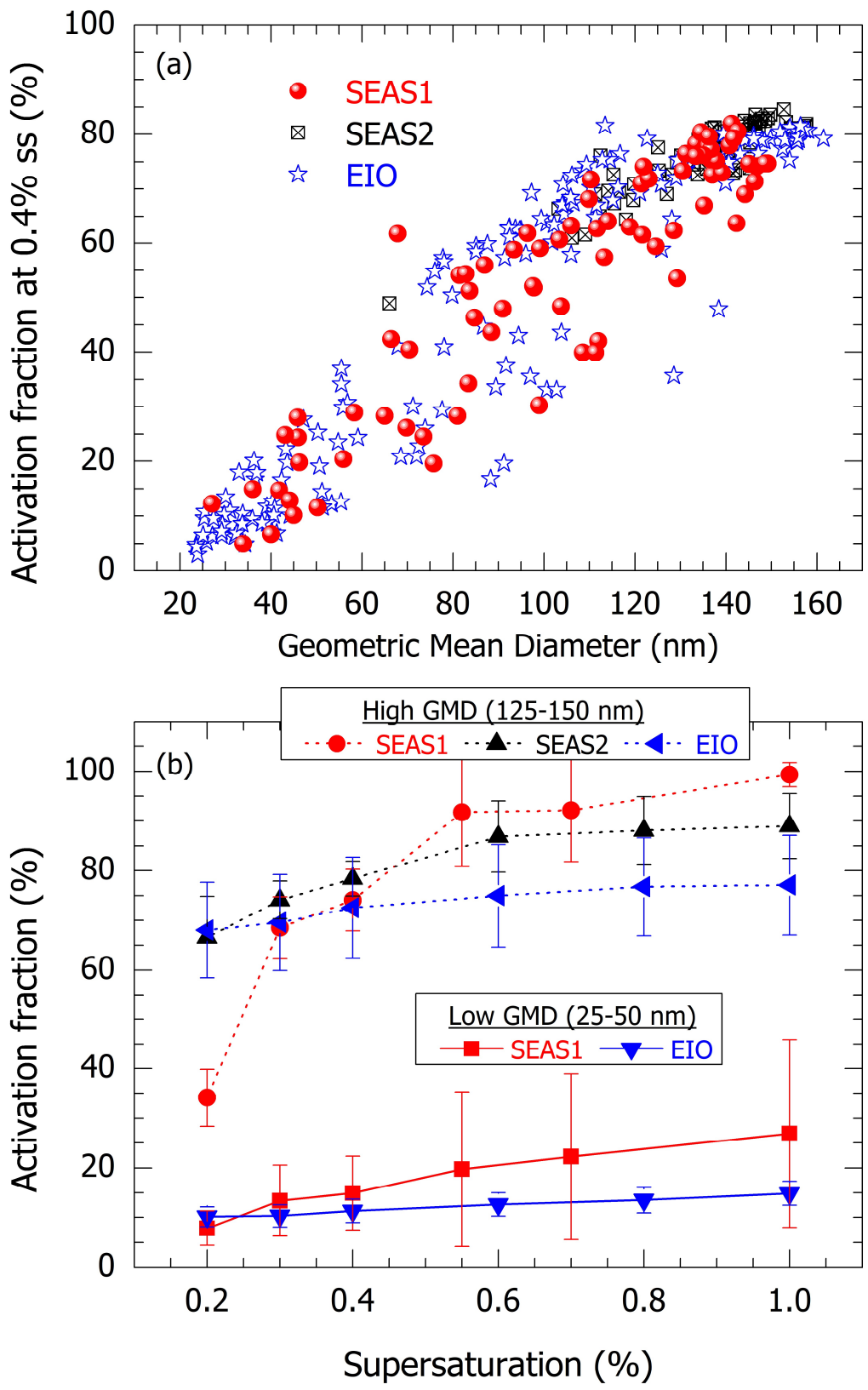

**Figure 6:** (a) Association of geometric mean diameter (GMD) with CCN activation fraction at 0.4% supersaturation over SEAS1, SEAS2 and EIO during ICARB-2018. (b) Supersaturation spectra of activation fraction for low (25-50 nm) and high (125-150 nm) GMD values during SEAS1 and EIO. All the size distribution measurements during SEAS2 have GMD above 50 nm. Vertical bars on the symbol indicate standard deviation of the measurements.

Above discussions highlighted the significant role of GMD, in turn, the abundance of ultrafine mode aerosols on the CCN activation efficiency of the aerosol system irrespective of the airmass origin and distance from the continental sources. This analysis broadly confirms the primary role of aerosol size distribution in deciding the CCN concentration and activation fraction over the region. During ICARB-2018, GMD values varied from 25 to 160 nm, and nearly 40% of the measurements have GMD less than 100 nm. The high activation values are observed whenever the contribution of ultrafine mode aerosols to the CN is negligible. Also higher CN values (>$10^4$ $cm^{-3}$) with low GMD values (<60 nm) are seen over both the SEAS1 and EIO regions suggesting new particle formation events. Low CCN efficiency at 0.2% supersaturation at SEAS1 reinstates the dominance of ultrafine particles or the hydrophobic nature of the particles resulting from the continental outflow. However, the presence of low CCN efficient aerosol system at higher supersaturation can be attributed to new particle

formation events at both the SEAS1 and EIO regions. It is well known that bimodal distribution is a characteristic of the aerosol system over marine environment (one mode emanating from possible nucleation events and the other one being omnipresent accumulation mode), whereas aged continental outflows over the oceans show unimodal distribution with mode around 80 to 200 nm (Yum et al., 2007). The measurements of aerosol particle size distributions carried out over the Bay of Bengal during winter 2009 (ICARB-2009) revealed a unimodal size distribution with mode diameter close to 100 nm. This further supports the lacks of ultrafine particles over SEAS2, where airmasses are mostly originating from the Bay of Bengal. At low supersaturations, marine aerosols have higher CCN activation fraction compared to the continental influenced aerosol system, which flip-flops at higher supersaturations. Present study also is in line with the earlier observations reported over various oceanic regions, which depicted higher activation fraction for continental aerosols than clean marine aerosols (Yum et al., 2007). A strong association is observed between GMD and hygroscopicity parameter for different mixing state conditions (external, internal and internal with non-growth) over Korean peninsula (Kim et al., 2018). Present study highlight that the higher activation fractions are observed for the continental or marine aerosols mostly in the absence of ultrafine particle bursts events.

### 3.3 Aerosol size distribution and CCN activation

To understand the effect of ultrafine particles on CCN activation, two typical cases have been considered over SEAS1 and EIO where aerosol size distribution varied drastically within a few hours due to ultrafine particle events which are characterized by the low GMD values and high number concentration of ultrafine particles (diameters < 100 nm). Figure 7 shows the CCN spectra and activation fraction along with the corresponding aerosol number size distribution over SEAS1 and EIO for high and low concentration of ultrafine particles (high-UFP and low-UFP). Corresponding k values (Twomey's empirical slope) are also mentioned in the figure. Over SEAS1, the variation of CCN with supersaturation for high and low UFP cases showed almost similar pattern with relatively steeper increase in CCN with supersaturation for high-UFP compared to low-UFP. The CCN activation efficiency for low-UFP case increased from ~33% at 0.2% supersaturation to almost 100% at 1.0% supersaturation, whereas for high-UFP the activation fraction at highest supersaturation (1.0%) is only ~45% over SEAS1. This large difference in the activation fraction for the two cases can mostly be attributed to the increase in ultrafine particles as already mentioned in the earlier section and Figure 7b. For the low-UFP case shown in Figure 7b, the particles below 100 nm contributed almost 37% to the total number concentration (CN) and all these particles are activated to CCN at high supersaturation in contrast to the high-UFP case. The large amounts of ultrafine particle concentrations (with GMD ~50 nm in Figure 7b) seen in the high-UFP case did not contribute significantly to the CCN concentrations even at 1.0% supersaturation.

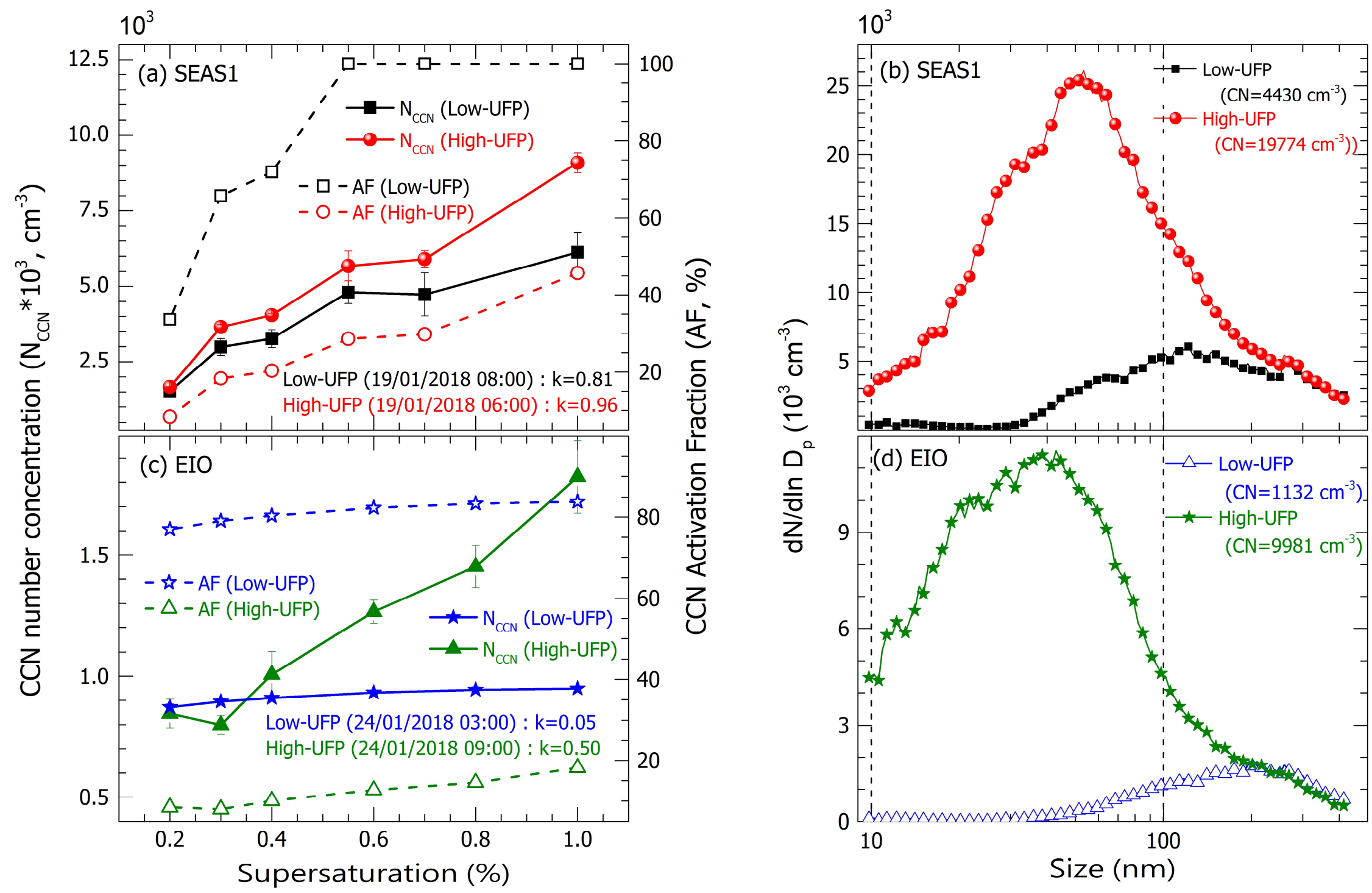

**Figure 7:** (a) & (c) Typical cases of supersaturation spectra of CCN number concentration and activation fraction (AF) values for high (case1) and low (case 2) concentration of ultrafine particles (UFP) over SEAS1 and EIO. (b) & (d) The aerosol number size distributions corresponding (a) and (c).

Over EIO, though the CCN values at lower supersaturations are comparable for both the cases, 100% increase is observed in CCN concentration at 1.0% supersaturation for high-UFP compared to low-UFP (Figure 7c). The activation fraction depicted an entirely distinct pattern (a flat spectra) with low values (20%) at all supersaturation for high-UFP compared to the activation fraction of 75% for low-UFP case. These typical events over SEAS1 and EIO clearly demonstrate the lower activation efficiency of ultrafine aerosols. However, there exist strong differences in the activation properties of CCN over SEAS1 and EIO. For both high and low-UFP cases, SEAS1 aerosols are more CCN active than EIO, which is clearly reflected in the regional mean values also. The absolute magnitude of CCN concentration is relatively less over EIO with no significant increase in CCN concentration with increasing supersaturation (especially for low-UFP, k ~0.05). This flat CCN spectra (lower k values <0.5 in both the cases) observed over EIO represent maritime aerosol system and high k values (>0.8) over the SEAS1 represent polluted marine conditions (Jayachandran et al., 2017). The aerosol system can have high and low activation fraction (k values) depending on the contribution of ultrafine mode aerosols to the total number concentration, which is in contrary to the general classification of the aerosol system based on the k values (Jayachandran et

al., 2017). The inverse relationship between Twomey exponent (k) and activation fraction reported by Jayachandran et al., (2017) over coastal location Thumba which is geographically closer to SEAS1 does not hold during ICARB-2018 because of the new particle formation events and abundance of ultrafine particles, similar to high-UFP cases as shown in Figure 7.

To evaluate the relative importance of aerosol number size distribution in CCN activation, similar aerosol size distributions are grouped by regressing the each individual size distribution with all the other size distributions. The size distribution having maximum number of occurrence of regression coefficient ($R^2$) greater than 0.9, is identified as first group. Further, the first group of distributions are removed from the dataset, and regression analysis is repeated to estimate the next major prominent size distribution. The mean number size distributions and corresponding mean activation fraction at 0.4% supersaturation for each of the groups are shown in Figure 8(b) (value written close to bar chart is number of observations). This analysis highlighted that majority of the hourly mean size distributions (number N~ 119) depicted a broad peak with mode around 150-300 nm (Type 1) with highest activation ratio of 76%. Whenever the mode of the size distribution falls below 100 nm, i.e., mode is in the ultrafine particle regime (Type 2, Type 5 and Type 6), activation fraction drops to a value below 20%. The Type 3 and Type 4 size distributions, which are similar to Type 1 but with a small difference in the particle concentration above 100 nm, also showed higher activation fraction (> 70%). The spread in activation fraction (standard deviation) for a 'fixed' aerosol size distribution imply the contribution of aerosol chemical composition. It should be kept in mind that, even though we grouped the similar distributions ($R^2 = 0.9$), the small variations observed between the member distributions (standard deviation of size distributions shown in Figure 8a) also can contribute to the standard deviation of the activation fraction. The difference between the activation fractions for two distinct size distributions is larger than their standard deviations, which initially implies the primary role of number size distribution on the CCN activation than the chemical composition.

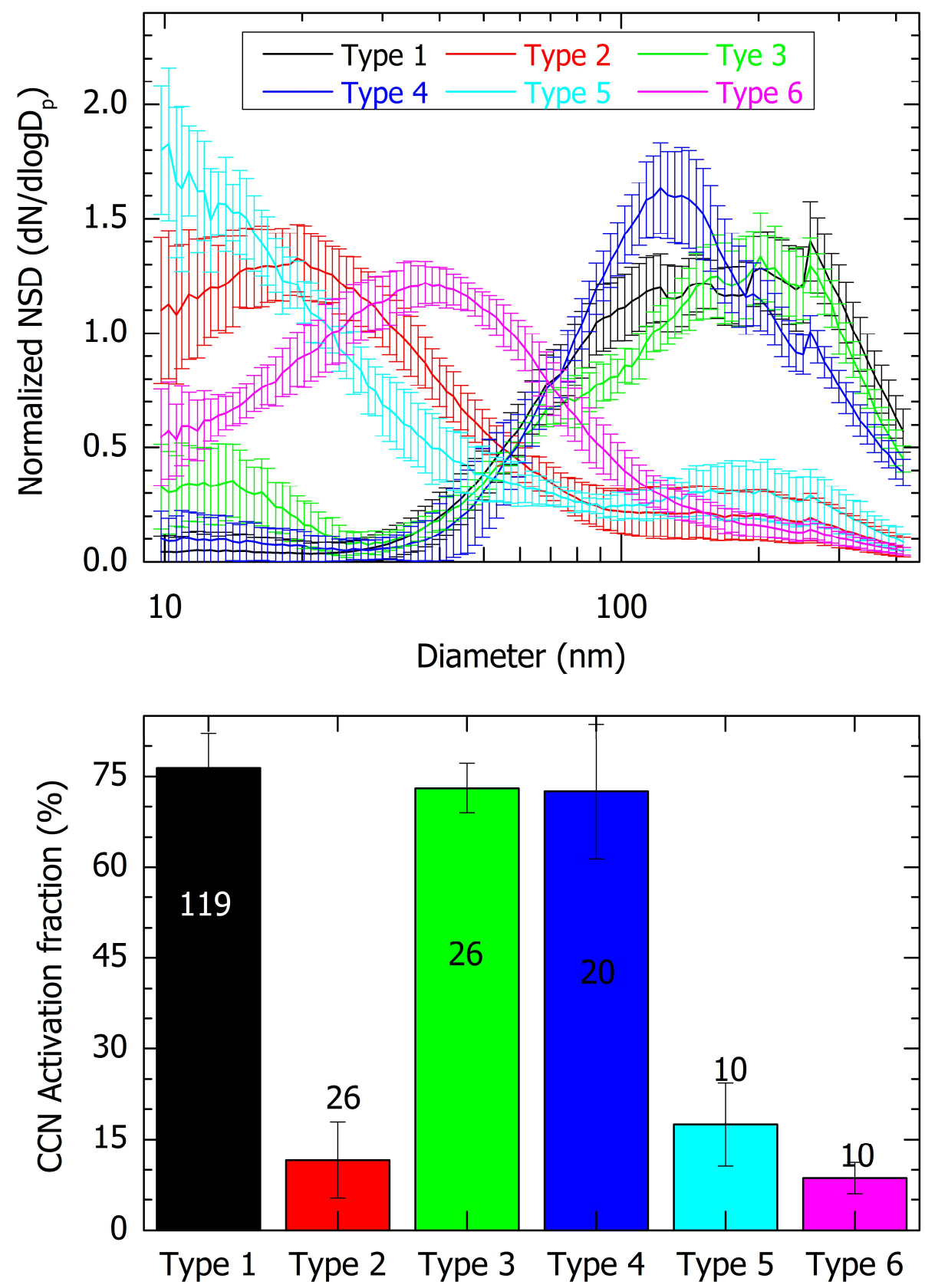

**Figure 8:** (a) Most frequent occurring aerosol number size distributions (normalized) during ICARB-2018. Vertical bars represent the standard deviation of the data and the number of hourly observations averaged for each distribution is given in bottom panel. (b) Activation fraction at 0.4% supersaturation of the normalized distributions shown in panel (a).

We have further examined the association of CCN concentration with aerosol number concentration (dN) at different size ranges. Figure 9a shows the scatter plot of CCN concentration with cumulative aerosol number concentration above the size range of 50 nm ($N_{50}$), 100 nm ($N_{100}$) and 150 nm ($N_{150}$). Slope of the regression fit and correlation coefficients estimated for the CCN and aerosol concentrations above different size ranges (up to $N_{250}$) are shown in Figure 9b. The slope of the regression analysis increased as lower size cut of the aerosol number concentration increased from 50 nm to 250 nm. The correlation coefficient was low for $N_{50}$ and which increased to 0.98 for $N_{100}$. The CCN varied linearly with CN where the slope of the variation strongly depend on the size distribution, and linearity is stronger for larger aerosols (GMD values >100 nm as shown in Figure 4) and weaker for ultrafine mode aerosols. The scatter between CCN at 0.4% supersaturation and $CN_{100}$ shows better association (slope ~ 0.95 and $R^2$ ~ 0.92). This implies that most of the particles above 100 nm get

activated as CCN at 0.4% supersaturation and $N_{100}$ is a good proxy for CCN concentration at 0.4% supersaturation. The contribution of particles to the CCN concentration is reduced drastically above 100 nm. For the particles above 200 nm, CCN concentration is nearly 2.5 times higher than the $N_{200}$.

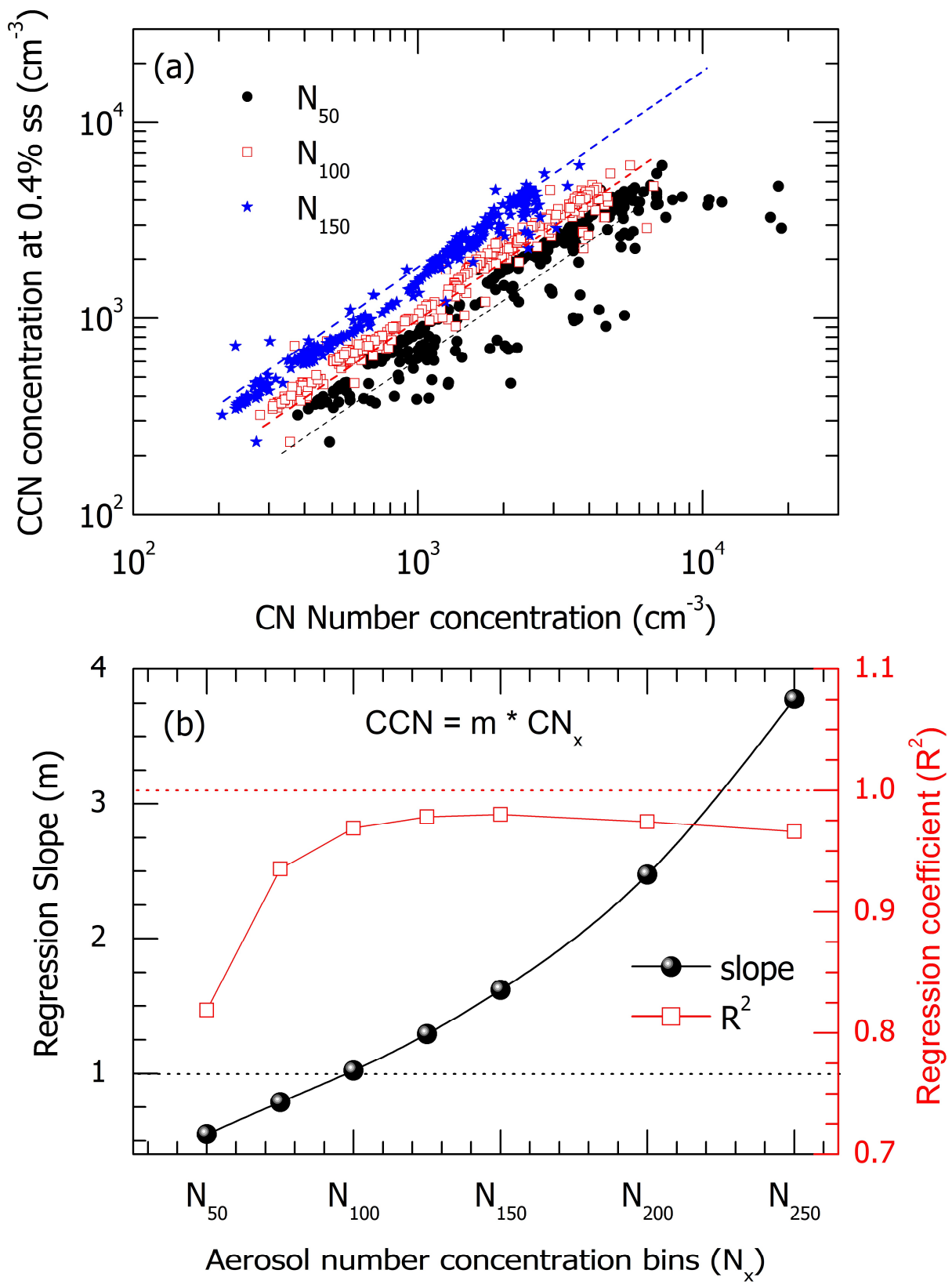

5 **Figure 9:** (a) Scatter plot between CCN number concentrations at 0.4% supersaturation with aerosol number concentration for particles size greater than 50, 100 and 150 nm. (b) Variation of regression coefficients (slope and $R^2$) estimated for CCN at 0.4% supersaturation with aerosol number concentration at different size ranges. The $N_{50}$ indicate aerosol number concentration above 50nm.

The above analysis indicates that the critical activation diameter, which is the size above which all the aerosols get activated as CCN is close to 100 nm for the continental outflow of aerosols to the northern Indian Ocean during wintertime. The critical activation diameter of the aerosol system can be experimentally estimated from the size segregated CCN measurements (Rose et al., 2010) or from the simultaneous observations of CCN and aerosol number size distributions (Furutani et al., 2008). Assuming an internally mixed aerosol system, we have estimated the 'apparent critical diameter' for a particular supersaturation by integrating the aerosol number concentration (dN) from higher to lower size range in such way

that estimated aerosol number concentration equals CCN concentration (Burkart et al., 2011) and the diameter corresponding to that is called critical diameter. The cumulative number concentration (summing from higher size to lower) and critical diameter for different supersaturations over SEAS1, SEAS2 and EIO are shown in Figure 10a and b. As seen in figure 10a, the contribution of ultrafine particles to the total concentration is relatively high over SEAS1 compared to SEAS2 and EIO due to the continental proximity. As expected, the critical diameter decreased with an increase in supersaturation for all the regions in consistency with the Kohler theory. Relatively large decrease in critical diameter with supersaturation was observed over SEAS1 compared to SEAS2 and EIO. As shown in Figure 2, though the aerosol loading is high over SEAS1, CCN concentration at 0.2% supersaturation is lower over SEAS1 compared to that of EIO. High value of critical diameter (~185 nm) over SEAS1 further confirm that only a small portion of the aerosol population is activated as CCN at 0.2% supersaturation. This is mostly attributed to the presence of less hygroscopic aerosols (primary carbonaceous) above 100 nm size range, which require higher supersaturation to get activated. In contrast, the lower critical diameter at 1.0% supersaturation over SEAS1 could be attributed to the water-soluble aerosol in the ultrafine particle mode. It is interesting to notice that the critical diameter at 1.0% supersaturation is higher for EIO than SEAS1 in contrast to 0.2% and 0.4% supersaturations. This observation is in line with the high activation fraction at 1.0% supersaturation over SEAS1 compared to EIO (Figure 6).

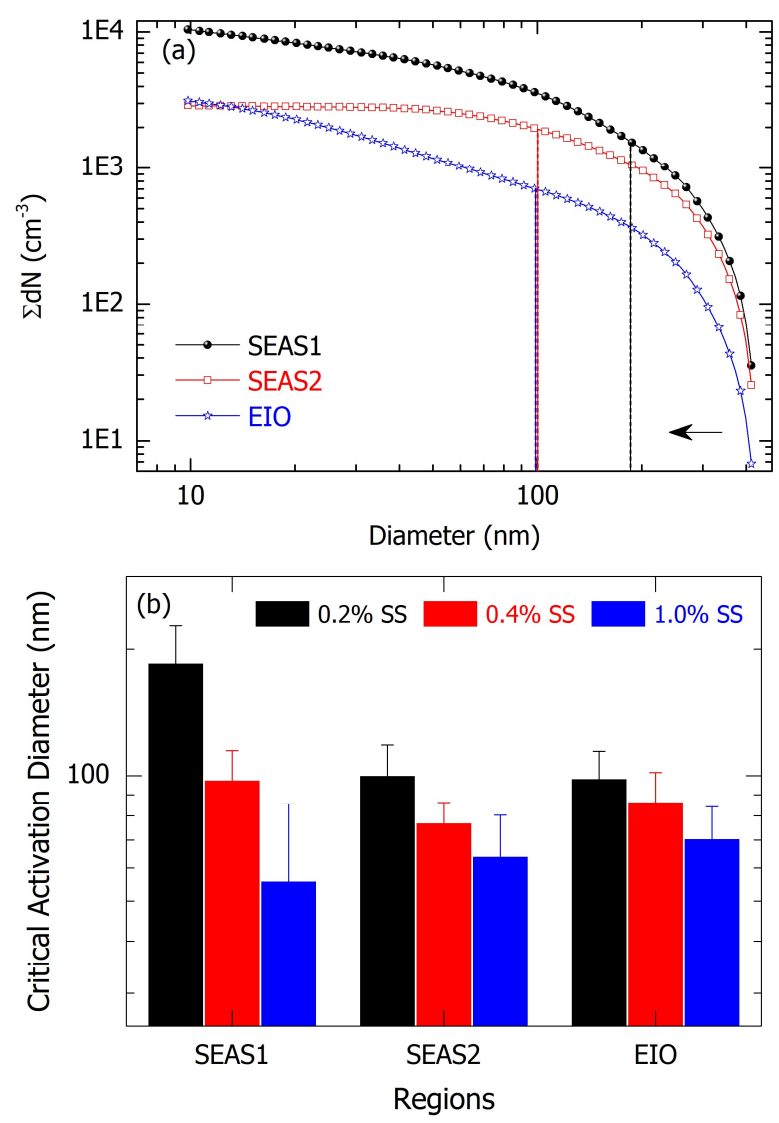

**Figure 10:** (a) Cumulative aerosol number concentration from higher size range to lower over SEAS1, SEAS2 and EIO. The value at 9 nm indicates the total number concentration. Apparent critical activation diameter is shown as dashed vertical lines. (b) Variation of critical diameter for 0.2%, 0.4% and 1.0% supersaturations over SEAS1, SEAS2 and EIO.

Simultaneous measurements aerosol number size distribution and CCN at different supersaturation is highly useful in understanding the CCN characteristics. The size distribution over SEAS1 has a mode close to 100 nm size range, which represents the influence of polluted continental airmass. Ueda et al., (2016) reported bi-modal size distributions as a typical maritime aerosol system based on the extensive measurements of aerosol number size distributions over Pacific Ocean covering 40°N to 40°S. However, most of the measurements during ICARB-2018 depicted broad mono-modal distribution indicating the aged, continental outflow aerosols and the less frequent bimodal distributions are mostly associated with ultrafine particle bursts (Figure 8). An accumulation mode has been consistently present in the aerosol number size distribution over SEAS, which is contributing to the high CCN activation. The regional difference in the aerosol number size distribution and its role in determining the activation fraction of CCN are vivid such that the dominance of fine/ultrafine particles decrease the CCN efficiency from the expected values. Kalivitis et al., (2015) investigated the association of CCN at 0.2% supersaturation with aerosol concentration larger than threshold diameter (from 80 to 130 nm) over eastern Mediterranean marine atmospheric boundary layer. These authors reported particle concentrations above 100 nm as a best indicator for CCN at 0.2% supersaturation over the region, and present study supports this finding (Figure 9). Burkart et al., (2011) attributed the high critical diameter of aerosols at 0.5% supersaturation over Vienna to the presence of insoluble aerosols. Rose et al., (2010) have reported dry activation diameter of 200 to 30 nm for a wide range of supersaturation 0.068 to 1.27%.

The presence of significant amounts of ultrafine particles observed over the northern Indian Ocean could be attributed to the in-situ new particle formation events and/or transport from the free troposphere. The ICARB-2018 measurements indicate that these ultrafine particle events are mostly observed during the early morning and evening hours. Nearly 20 to 40% increase in the mass concentration of organic carbon (OC) was observed during such conditions, implying the formation of secondary organic aerosols. Large scale anti-cyclonic system prevailed during the wintertime over the region also support the amalgamation of distinct airmasses and intrusion of ultrafine particles formed in the free troposphere towards the lower atmosphere. The absence of any open mode towards the lower size regime during such ultrafine particle bursts and lack of extremely high number concentration at10 nm size bin (lowest in the present SMPS setup) further confirms the possibility of the transport of free tropospheric aerosols. Nair et al., (2013) have reported ultrafine particle dominance over the northern Arabian Sea during the spring 2006, which was strongly associated with the variation of chlorophyll concentrations implying the role of ocean biogeochemistry. However, the role of ocean biogeochemistry and that of the semi-volatile organic vapours in the formation of ultrafine particle in the present study needs to be investigated in detail.

Since the CCN activity depends mainly on the aerosol size and composition (Dusek et al., 2006a) critical diameter can be considered as a proxy for the variations in the chemical composition of the aerosol system. It should be reinstated that the aerosol size distribution and chemical composition are intrinsically coupled with each other and any change in the aerosol size distribution may have change in the aerosol composition as well. Quinn et al., (2008) have correlated the critical diameter with the hydrogenated organics aerosols (HOA) mass concentration, and found that HOA can explain about 40% of

the variance in the critical diameter. For anthropogenic and marine environments, Furutani et al., (2008) have reported critical diameter of 70-110 nm and 50-60 nm, respectively, at 0.6% supersaturation. The critical diameter estimated for the South Asian outflow is comparable to the values of 70-90 nm (at 0.44% supersaturation) reported by the Quinn et al., (2008) for marine regions. It should be noted that the freshly emitted carbonaceous combustion particles have high critical diameter (~350 nm) even at a high supersaturation (0.7%) as reported by Dusek et al., (2006b). The presence of soluble aerosols largely enhance the CCN activity of insoluble particles such as BC and dust (Dusek et al., 2006b) due to effective mixing. Thus, these inferences underline the need for more-realistic observations of mixing state and size-segregated aerosol composition measurements at regions like SEAS, where primary as well as secondary particles are present from both natural and anthropogenic sources.

This study highlighted the high concentration of CCN over the South Asian outflow regions. The relative importance of aerosol size distribution and chemical composition on CCN activation has been the topic of investigation for several experiments (Dusek et al., 2006a; Kerminan et al., 2012). Significant understanding on the critical diameter (Kerminan et al., 2012) and size segregated hygroscopicity parameter (κ) (Petters and Kreidenweis, 2007) enabled us to predict the CCN concentration based on the measurement of physical and chemical properties of aerosols carried out at distinct environments (Schmale et al., 2018). Paramonov et al., (2015) have showed that hygroscopicity parameter decreases with particle size, with the Aitken and accumulation mode aerosols having statistically significant difference in κ values. Our observations with the lower activation fraction observed for distributions with ultrafine particle dominance are in concurrence with Paramonov et al., (2015). Several studies have considered an average size distribution of aerosols with varying hygroscopicity parameter to predict CCN concentration (Rose et al., 2010, Gunthe et al., 2009; Jurányi et al., 2010). These studies also emphasised that aerosol size distribution has major role in deciding the number concentration of CCN. Meng et al., (2014) reported that hygroscopicity is more important at low supersaturations and vice versa at high supersaturations. While the present study described the CCN characteristics over the different parts of the northern Indian Ocean and its association with particle number size distributions, further studies are required to examine the climate implications of these observations.

## 4. Conclusions

Extensive measurements of the aerosol and CCN properties in South Asian outflow to the northern Indian Ocean were carried out as a part of ICARB-2018 experiment during January-February 2018. The influence of continental outflow on the CCN characteristics over the marine atmospheric boundary layer extending from 15°N to 2°S (Southeast Arabian Sea and equatorial Indian Ocean) close to the Indian sub-continent were investigated. The major highlights of this study are

- High CCN concentration are seen over Southeastern Arabian Sea, with a steep (k=0.83) CCN spectra, while low values (higher than previously reported pristine marine) are observed at the equatorial Indian Ocean.
- At high (1.0%) supersaturations, most of the aerosols in the south Asian outflow (over SEAS) get activated as CCN, whereas aerosol system over the equatorial Indian Ocean is less CCN efficient even at higher supersaturations.

- The CCN efficiency depicts a strong association with geometrical mean diameter of the aerosol number size distribution. The activation efficiency decrease with the dominance of ultrafine particles in the size distribution.
- Accumulation mode particles (> 100 nm) contribute to the high activation fraction (69%) over the northern Indian Ocean. The number concentration of particles above 100 nm is a good proxy for predicting CCN number concentration at 0.4% supersaturation.

The formation and transport pathways of ultrafine particles over the region during the winter season remains an open question at present and more dedicated field experiments and detailed investigations are required to address this in detail. Though the variations in GMD of the aerosol number size distributions accounts for the variability in CCN activation efficiency over the Southeastern Arabian Sea and the equatorial Indian Ocean, the change in aerosol chemistry associated with the ultrafine particle burst events during the early morning and evening hours needs further investigation using the size segregated CCN measurements and online measurements of aerosol chemistry.

## Acknowledgements

The ICARB-2018 experiment was carried out under the ISRO Geosphere Biosphere Programme. Authors acknowledge the National Centre for Polar and Ocean Research (NCPOR) of Ministry of Earth Sciences for providing the ship board facilities onboard ORV Sagar Kanya. We acknowledge NOAA ARL for the providing the Hybrid Single-Particle Lagrangian Integrated Trajectory (HYSPLIT) transport and dispersion model used in this study.

## Author contributions

SSB and VSN designed the experiment. VSN, JV, SKK, MMG and SSB involved in the data collection on-board ship. VSN did the scientific analysis of the data and drafted the manuscript. SSB edited the manuscript.

## Data availability

ICARB-2018 data are available upon request from the contact author, Vijayakumar S. Nair (vijayakumar_s@vssc.gov.in).

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
