# Peer review of "Cloud Condensation Nuclei properties of South Asian outflow over the northern Indian Ocean during winter"

_Atmospheric Chemistry and Physics, 2019_

## Referee Comment (RC1) · James Hudson (Referee) · 27 Nov 2019

This manuscript is worthy of publication after some revisions.

The term ultrafine is not incorrect, but it is misleading. It begs the question of the fine mode (100-2500 nm), which is not mentioned, but which is synonymous with accumulation mode, which is mentioned. The coarse mode is mentioned but I do not see any measurements larger than 2500 nm (2.5 µm) or even 1µm by some fine mode definitions. It is more appropriate to use Aitken rather than ultrafine when comparing with accumulation.

Figs. 3, 7a, 7c. CCN spectra are generally and traditionally plotted log-log so that the slope, k, is easily illustrated. Furthermore, when presenting k, the supersaturation range must be specified.

Analysis of Fig. 3 indicates that k was calculated between 1 and 0.2%. This is rather deceptive for the red (SEAS1) and black (SEAS2) data where k varies over S. The 0.83 k is thus exaggerated compared to 0.53 for green. Furthermore, it is also necessary to tell all readers that these are cumulative concentrations, all particles with critical S less than the specified S. Thus, concentrations must always increase or remain the same toward the right of such figures. This is certainly not the case for green (especially solid) in Fig. 7c between 0.2 and 0.3% and Fig. 7a for solid black between 0.6 to 0.7% and possibly black between 0.8 and 1% of Fig. 3. This can of course be due to experimental uncertainty/error. But the omission of noting cumulative concentrations could mislead readers less familiar with CCN. This is especially inconsistent with the later noting of cumulative concentrations in Figs. 4 & 10. This issue needs discussion.

P6. L13-14. The concentration did not increase. Increase implies change over time. The cumulative concentration is merely higher at higher S as it should be. It is a lot greater at higher S (higher k) in SEAS1 compared to SEAS2 and EIO where concentration is more similar with S.

P7. L15. Why? Possibly due to cloud processing in maritime clouds (Hoppel et al., 1985, 1986, 1990, 1994, 1996; Hudson et al., 2015, 2018) ?

P8. L 5&6. Actually, more like 1200 (1190) and 200 (176) cm$^{-3}$. Also Fig. 2 of that paper reveals concentrations at the various S used in this manuscript for comparison. It demonstrates similar low k for cleaner air but not such high k of polluted air.

L20. Hudson & Yum (2002) Fig. 1 show the latitudinal decrease of CCN and CN. This continental influence ceased below 5° south latitude due to the intertropical convergence zone.

Fig. 4a. The mixture of open and closed symbols in the legend is misleading as is putting colors into the legend. The colors only refer to the GMD color scale at the right. All symbols should be open so that the overlapping data can be discerned. This figure is difficult enough to understand even without these distractions.

Fig. 4b. These data could be plotted against each other to better reveal their relationship/correlation.

Fig. 5. Fraction and percentage are confused.

Fig. 9a. Symbols should be open. Again, the mixture of open and closed legend symbols is confusing.

P10. L12. C was used on p7L6, $N = CS^k$. C here is entirely different and should be represented by a different symbol.

P11. L16. How do you know that this is irrespective of chemistry.

P12. L2. How do you know that this is irrespective of chemistry.

L14-15. I do not see this. Fig. 6 considers only 0.4%.

Fig. 9b. $R^2$ is coefficient of determination. R is correlation coefficient.

P13. L7-8. This does not make sense.

P14. L3. The accumulation mode is not omnipresent (Hoppel et al. & Hudson et al.) It is caused by cloud processing (Noble & Hudson 2019).

P16. L5. Explain this regression.

L6. $R^2$ is not the regression coefficient, it is the coefficient of determination.

P20. L17-18. More likely just continental pollution.

P20 L23. Explain open mode.

P21 L12. Also Hudson & Da (1996) and Hudson (2007).

P21 L30-31. How can such low ks be higher than previous measurements? Fig. 2 of Hudson & Yum (2002) shows k 0.23 for 1-0.2% S.

P21 L32. They do not get activated as CCN. They are CCN.

L33. This would imply that there are many CN that are not CCN, but I do not see this in this manuscript.

Minor:

Fig. 2. It would be more appropriate to reverse the positions of 0.2% and 1% in the legend.

P2. L12. Insert The before South.

L13. In to within.

L14. Delete the.

L23. Add Hudson & Yum (2002).

L28. Change never to not.

P3. L1-2. Add Hudson (2007).

L4. Delete the.

L5. Insert a before few. Level plural.

L15. Delete 2nd the.

P4. L12.

P5. L32. Delete enough.

P6. L1. Change values to concentrations.

L4-5. Delete in the.

L9. Delete in.

L12. Delete 1st the.

L13-14. Change increased by to was. Folds singular. Change when to higher at. Move 1% before supersaturation. Replace changed from to than at. Delete to. Change increase to concentration was similar. Delete is significant.

P7 L1. Delete as.

L5. Delete 's. Add ship to relation.

L7. Add s to indicate.

L13. Insert somewhat before hydrophobic. Hydrophobic particles would not be CCN at all.

P8. L18. Delete s of towards.

P9. L12. Delete has.

L13. Is to was.

L17. Change get activated to CCN.

L18. Delete last the.

P10. L16. Insert for after accounting.

L18. Define coarse mode.

L20.  Particles singular.

L21.  Insert the after in.

P11. L2.  Distribution plural.

L7.  Insert the after from.

L12.  Insert in EIO after particles.  Change ed to s.

L13.  Insert concentration after CN.

L23.  Pattern plural.

P12.  L7.  Distribution plural.

L14-15.  This implies relatively higher CN at EIO.

P13. L8.  Activation efficiency is redundant with CCN.  It is not necessary.

P14. L13.  Explain flip-flops.

L13.  Bursts singular.

L17.  Delete the.

L23.  This is so even at 0.55%.

L24.  It is not an increase.  It is higher concentrations.

L26.  This is not shown.

L26-7.  Delete activated to.  This is redundant.  If they are CCN they can be activated.

P15. L8.  Change spectra to distribution.  Spectra implies CCN S spectra.  Delete a.
Supersaturation plural.

L11.  Not so for 0.2 & 0.3%.

L 12.  Move also before clearly.

L14.  Delete system and then end the sentence.

L15.  Delete 1$^{st}$ the.

L16.  Delete parentheses and values.  Insert or before k.

L17.  Period after concentration.  Change which to This.  Delete in.  delete last the.
Delete values.

P16. L2.  Comma after Thumba.

L3.  Period after particles.  Insert This is.

L5. Delete 1$^{st}$ the.

L7.  Change Further to Then.  Delete from the data set.

L8.  Fig. 8a. is not mentioned.

L12-13.  Activation fraction does not drop.  It is lower.

P17. L11.  Coefficient of determination.  Delete which.

L12.  Add ed to depend.

P18.  L1.  Change activated as to are.

L2.  Insert at 0.4% after concentration.  Change reduced drastically to lower.

L10-11.  Specify S, probably 0.4%.

L12.  Delete last the.

P19. L5.  Change decreased with an increase in to was lower for higher.  Supersaturation plural.

L6.  Period after regions.  Change in to This is.  Change consistency to consistent.  Delete
the.

L10.  Change primarily to possibly.

P20. L1.  Insert of before aerosol.  Delete highly.

L2.  Delete 1$^{st}$ the.  Delete size range.

L4.  Move the after over.

L28.  Particle plural.

L31.  Change reinstated to emphasized.
L33.  Change change in the aerosol to also been due to.  Change as well to differences.
P21. L12.  Insert Hudson (2007).
L20.  Insert the before major.
L21.  Change vice versa to not.
L26.  Delete 1st the.
L30.  Concentration plural.  Insert k after low.
L32.  Move at high (1.0%) supersaturations after CCN.  Change get activated as to are.
L33.  Delete even.
P22.  L2.  Change 2nd the to greater.  Delete in the size distribution.

Hoppel, W.A., Fitzgerald, J.W., & Larson, R.E. (1985).  Aerosol size distributions in air masses advecting off the East Coast of the United States.  *Journal of Geophysical Research,* 90, 2365-2379.

Hoppel, W.A., Frick, G.M., & Fitzgerald, J.W. (1996).  Deducing droplet concentration and supersaturation in marine boundary layer clouds from surface aerosol measurements.  *Journal of Geophysical Research,* 101, 26,553-26,565.

Hoppel, W.A., Frick, G.M., Fitzgerald, J.W., & Larson, R.E. (1994).  Marine boundary layer measurements of new particle formation and the effects nonprecipitating clouds have on aerosol size distribution.  *Journal of Geophysical Research,* 99, 14443–14459, doi:10.1029/94JD00797, (H94).

Hoppel, W.A., Frick, G.M. & Larson, R.E. (1986).  Effect of nonprecipitating clouds on the aerosol size distribution in the marine boundary layer.  *Geophysical Research Letters,* 13, 125-128.

Hoppel, W.A., Fitzgerald, J.W., Frick, G.M., Larson, R.E., & Mack, E.J. (1990).  Aerosol size distributions and optical properties found in the marine boundary layer over the Atlantic Ocean.  *Journal of Geophysical Research,* 95**,** 3659-3686.

Hudson, J.G., 2007: Variability of the relationship between particle size and cloud-nucleating ability.  *Geophys. Res. Let*., **34**, L08801, doi:10.1029/2006GL028850.

Hudson, J.G. and X. Da, 1996:  Volatility and size of cloud condensation nuclei.  *J. Geophys. Res., 101*, 4435-4442.

Hudson, J.G., Noble, S., & Tabor, S. (2015).  Cloud supersaturations from CCN spectra Hoppel minima.  *Journal of Geophysical Research-Atmospheres*, 120, 3436–3452, doi:10.1002/2014JD022669.

Hudson, J.G., S. Noble, and S. Tabor, 2018: CCN spectral shape and stratus cloud and drizzle microphysics.  *Journal of Geophysical Research: Atmospheres,* 123, 9635-9651. http://doi.org/10.1029/2017JD027865

Noble, S.R., and Hudson, J.G., 2019: Effects of continental clouds on surface Aitken and accumulation modes.  *Journal of Geophysical Research: Atmospheres,* **124**, 5479-5502. https://doi.org/10.1029/2019JD030297.

---

## Referee Comment (RC2) · Anonymous Referee #2 · 8 Jan 2020

Review of "Cloud Condensation Nuclei properties of South Asian outflow over the northern Indian Ocean during winter" by Vijayakumar S Nair et al., (ACP-2019-828)

This paper presents some interesting results on cloud condensation nuclei (CCN) and condensation nuclei (CN) concentrations obtained over north Indian ocean as a part of ICARB-2018 conducted during winter 2018. Sophisticated data collected within the ship cruise campaign (16 Jan. 2018 to 13 Feb.2018) has been wisely used to investigate the latitudinal and longitudinal variations. Major conclusions drawn from this detailed investigations includes findings of high CCN over southeastern Arabian sea compared to equatorial Indian ocean, high CCN efficiency over south Asian outflow

compared to equatorial Indian ocean, contribution of accumulation mode to high activation fraction over north Indian ocean. Further, strong association between CCN efficiency and geometrical mean diameter of aerosol number size distribution is being reported.

In general, paper is concise and well written with substantial new information and apt for Atmospheric Chemistry and Physics Journal. However, few clarifications are required before accepting for its publication. Below are the some of the issues which authors need to take care. Authors are strongly encouraged to revise this manuscript.

Comments/Suggestions:

Page 3, Lines 28-31, Figure 1:

The spatial extend of the aerosol transport to the Indian Ocean is qualitatively depicted by the climatological (2002-2017) mean aerosol optical depth (AOD) derived from MODIS observations over the northern Indian Ocean (contours in Figure 1). These contours are hard to see from the figure. I suggest including color contour of the same.

Page 4, Lines 5-7:

The first phase of the cruise over southeastern Arabian Sea (SEAS) is divided into 'SEAS1' and 'SEAS2' regions where the former is influenced by the air masses from peninsular India, and the latter is from the Bay of Bengal. I wonder how the lines are drawn that differentiates SEAS1 and SEAS1.

Page 4, Lines 11-12:

Indian sub-continent during this period due to the. This is an incomplete sentence.

Page 17, Figure 8:

I am unable to see much difference between Type 3 and Type 4. Did it qualify the followed analysis procedure?

Page 20, Line 3:

Reported 'bi-modal' size distributions should be reported 'bi-model' size distributions.

There are few grammatical mistakes and typos. I suggest authors to go through the manuscript carefully again before submitting revised draft.

—END—

---

## Author Comment (AC1) · 28 Jan 2020

*Review of Nair et al. This manuscript is worthy of publication after some revisions.*
**We thank the reviewer for very extensive and highly encouraging comments.**

*The term ultrafine is not incorrect, but it is misleading. It begs the question of the fine mode (100-2500 nm), which is not mentioned, but which is synonymous with accumulation mode, which is mentioned. The coarse mode is mentioned but I do not see any measurements larger than 2500 nm (2.5 μm) or even 1μm by some fine mode definitions. It is more appropriate to use Aitken rather than ultrafine when comparing with accumulation.*
**We thank the reviewer for this suggestion.**
**There are different nomenclatures used in the literature to estimate the aerosol concentration from the size segregated measurements. The term ultrafine is widely used in the literature to refer the particles below 100 nm (Pierce and Adams, 2007, Kumar et al., 2014; Hara et al., 2011). However, there are some studies considered ultrafine as a synonym to nucleation mode also (Kulmala et al., 2004). So, we have defined the term ultrafine as particles size less than 100 nm and fine as particles above 100 nm size in the revised manuscript. Since we are not reporting the coarse mode aerosol properties in this study, the term accumulation and coarse mode" is replaced with "fine mode" in the revised manuscript.**

**Kumar et al., (2014) Ultrafine particles in cities, Environmental International, 66, 1-10.**
**Kulmala et al., (2004) Formation and growth rates of ultrafine atmospheric particles: a review of observations, Journal of Aerosol Science, 35(2), 143-176.**
**Hara et al, (2011) Seasonal features of ultrafine particle volatility in the coastal Antarctic troposphere, Atmos. Chem. Phys., 11, 9803–9812.**

*Figs. 3, 7a, 7c. CCN spectra are generally and traditionally plotted log-log so that the slope, k, is easily illustrated. Furthermore, when presenting k, the supersaturation range must be specified.*
**Complied with. Figure 3 and 7 are changed to log-log scale.**

[Figure]

**Figure 3.**

[Figure]

**Figure 7:**

**The k values show dependence on the range of the supersaturation used for its calculation. As suggested by the reviewer, we have explicitly mentioned the supersaturation range used for the estimation of k value in the revised manuscript**

as **"In this study, k values are estimated in the supersaturation range from 0.2% to 1.0%."**

*Analysis of Fig. 3 indicates that k was calculated between 1 and 0.2%. This is rather deceptive for the red (SEAS1) and black (SEAS2) data where k varies over S. The 0.83 k is thus exaggerated compared to 0.53 for green. Furthermore, it is also necessary to tell all readers that these are cumulative concentrations, all particles with critical S less than the specified S. Thus, concentrations must always increase or remain the same toward the right of such figures. This is certainly not the case for green (especially solid) in Fig. 7c between 0.2 and 0.3% and Fig. 7a for solid black between 0.6 to 0.7% and possibly black between 0.8 and 1% of Fig. 3. This can of course be due to experimental uncertainty/error. But the omission of noting cumulative concentrations could mislead readers less familiar with CCN. This is especially inconsistent with the later noting of cumulative concentrations in Figs. 4 & 10. This issue needs discussion.*
**We agree with the reviewer.**

**The variation of CCN concentration with respect to supersaturation, represented by k value, is different over the three different regions (SEAS1, SEAS2 and EIO) under investigation. As reviewer suggested, we have mentioned explicitly that "As the set supersaturation inside the instrument increases, more and more particles will get activated since the supersaturation inside the instrument column is higher than the minimum supersaturation (called critical supersaturation) required for the particles to get activated. Hence CCN at higher supersaturation are cumulative and all particles with critical supersaturation lower than the set supersaturation are activated as CCN." [Page No. 7]**

**As reviewer correctly pointed out, the slight decrease in CCN at higher supersaturation compared to the CCN at nearest low supersaturation is mostly due to the experimental uncertainty or instrument error. When the aerosol size distribution is highly dynamic and changes within the time required to complete the CCN measurements at 5 supersaturations (ie 30 minutes), the supersaturation spectra of CCN will deviate from the Twomey's empirical fit. "In general, the CCN concentration will be higher or remain constant as supersaturation increases."**

*P6. L13-14. The concentration did not increase. Increase implies change over time. The cumulative concentration is merely higher at higher S as it should be. It is a lot greater at higher S (higher k) in SEAS1 compared to SEAS2 and EIO where concentration is more similar with S.*
**We have modified the sentence in the revised manuscript as "The CCN concentration at 1.0 % supersaturation is 5 to 7 folds higher than that at 0.2 % supersaturation over SEAS1, whereas the rate of change of CCN concentration with supersaturation is insignificant over SEAS2 and the equatorial Indian Ocean."**

*P7. L15. Why? Possibly due to cloud processing in maritime clouds (Hoppel et al., 1985, 1986, 1990, 1994, 1996; Hudson et al., 2015, 2018)?*
**We thank the reviewer for this suggestion. This aspect is included in the discussion. "The low k values observed over the SEAS2 and EIO also could be due to the cloud**

processing of maritime clouds (Noble and Hudson, 2019; Hudson et al., 2015)."
[Page No. 9]

*P8. L 5&6. Actually, more like 1200 (1190) and 200 (176) cm-3. Also Fig. 2 of that paper reveals concentrations at the various S used in this manuscript for comparison. It demonstrates similar low k for cleaner air but not such high k of polluted air.*
**We have included the correct values given in Hudson and Yum (2002). "Similarly, airborne measurements on board research flight (NCAR C-130) during INDOEX reported the values of CCN at 1.0% supersaturation as ~1190 cm$^{-3}$ for polluted airmasses and less than 176 cm$^{-3}$ for clean marine conditions over the southern Indian Ocean (Hudson and Yum, 2002)."**

**The k value reported from ICARB-2018 measurements over the equatorial Indian Ocean matches with INDOEX measurements (Hudson and Yum, 2002).**

**The high k values are observed over southeastern Arabian Sea (SEAS1, north of 7°N), which is close to the Indian peninsula, during ICARB-2018. INDOEX measurements did not show high k values, since those measurements were limited to south of 4°N (Hudson and Yum, 2002) and airmass back-trajectories were used to identify the polluted airmass. As seen in the ICARB-2018, the CCN concentration and k values generally decrease as we move towards the open ocean.**

*L20. Hudson & Yum (2002) Fig. 1 show the latitudinal decrease of CCN and CN. This continental influence ceased below 5° south latitude due to the intertropical convergence zone.*
**Complied with. The role of ITCZ on the latitudinal gradient of CCN is included in the revised manuscript as "Hudson and Yum (2002) reported that the influence of pollution outflow from South Asia to Indian Ocean ceased at 5°S due to the intertropical convergence zone."**

*Fig. 4a. The mixture of open and closed symbols in the legend is misleading as is putting colors into the legend. The colors only refer to the GMD color scale at the right. All symbols should be open so that the overlapping data can be discerned. This figure is difficult enough to understand even without these distractions.*
**Thanks for the suggestion. Figure 4 is modified as suggested.**

[Figure]

*Fig. 4b. These data could be plotted against each other to better reveal their relationship/correlation.*

**Figure 4 is modified as suggested.**

[Figure]

*Fig. 5. Fraction and percentage are confused.*

**Thanks for the suggestion. We understand the concern raised by the reviewer. We have replaced "activation fraction" with "activation efficiency" throughout the manuscript.**

[Figure]

*Fig. 9a. Symbols should be open. Again, the mixture of open and closed legend symbols is confusing.*
**Complied with.**

[Figure]

*P10. L12. C was used on p7L6, N = CSk. C here is entirely different and should be represented by a different symbol.*

**Sorry for that mistake. We have replaced 'C' with 'n'. "The regression coefficient between measured and estimated CCN further increased to 0.94 for a power law, $CCN_{est} = n*CN*GMD^p$ where p (1.5) and n (2000) are constants estimated iteratively for the highest value of $R^2$."**

*P11. L16. How do you know that this is irrespective of chemistry.*

**Since the effects of aerosol chemical composition on CCN activation were not addressed in the present study, we have removed that sentence.**

*P12. L2. How do you know that this is irrespective of chemistry.*

**Since the effects of aerosol chemical composition on CCN activation were not addressed in the present study, we have modified the sentence in the revised manuscript. By considering the aerosol size (GMD) information and total aerosol number concentration, we could predict CCN number (Figure 4). This is further supported by the decrease in activation efficiency (Figure 8) when the mode of the number size distribution changes from fine to ultrafine size range. However, change in aerosol size distribution inherently associated with a change in chemical composition and hence it is very difficult to delineate the effect of aerosol size and chemistry on activation efficiency.**

*L14-15. I do not see this. Fig. 6 considers only 0.4%.*

**Fig. 6b shows the CCN variation for different supersaturations.**

[Figure]

*Fig. 9b. R2 is coefficient of determination. R is correlation coefficient.*
**Complied with.**

[Figure]

*P13. L7-8. This does not make sense.*

**We have modified the sentence. "Above discussions highlighted the significant role of GMD, in turn, the abundance of ultrafine aerosols on the activation efficiency of the aerosol system."**

*P14. L3. The accumulation mode is not omnipresent (Hoppel et al. & Hudson et al.) It is caused by cloud processing (Noble & Hudson 2019).*

**We agree with the reviewer. We have removed the term "omnipresent" in the revised manuscript. In all the measurements made during INDOEX, ICARB-2006, ICARB-2009 and ICARB-2018, a prominent mode at 100 nm was observed over the oceans surrounding the Indian peninsula (Nair et al., 2013). Similar, mode was observed over coastal station Trivandrum and IGP outflow region Bhubaneswar (Babu et al., 2016). There are several studies have pointed out that this mode is originated from the cloud processing. Our intention was to just mention that this**

mode was observed during most of the campaigns (Hudson et al., 2015; Noble and Hudson, 2019).

Babu, S. S., S. K. Kompalli and K. K. Moorthy (2016) Aerosol number size distributions over a coastal semi urban location: Seasonal changes and ultrafine particle bursts, Science of the total Environment, 563, 351-365.

Hudson, J.G., Noble, S., & Tabor, S. (2015). Cloud supersaturations from CCN spectra Hoppel minima. Journal of Geophysical Research-Atmospheres, 120, 3436–3452.

Noble, S.R., and Hudson, J.G., 2019: Effects of continental clouds on surface Aitken and accumulation modes. Journal of Geophysical Research: Atmospheres, 124, 5479-5502.https://doi.org/10.1029/2019JD030297.

*P16. L5. Explain this regression. L6. R2 is not the regression coefficient, it is the coefficient of determination.*
**Complied with.**

*P20. L17-18. More likely just continental pollution.*
**Influence of South Asian outflow to the Indian Ocean can be seen up to ITCZ (Hudson and Yum, 2002). These authors reported a systematic decrease in aerosol loading towards ITCZ. Similar latitudinal gradient in black carbon mass concentration and columnar aerosol optical depth was reported by several authors (Nair et al., 2018). The number concentration of ultrafine concentration during ICARB-2018 did not show a systematic latitudinal gradient, which is mostly attributed to the limited number of ultrafine particle events and more events are observed over equatorial Indian Ocean than the southeastern Arabian Sea. So, we infer that these particles may not be directly transported from the continent rather nucleated from the low-volatile precursors and then transported.**

Nair, V. S., S. S. Babu, and K. K. Moorthy (2008), Aerosol characteristics in the marine atmospheric boundary layer over the Bay of Bengal and Arabian Sea during ICARB: Spatial distribution and latitudinal and longitudinal gradients, J. Geophys. Res., 113, D15208, doi:10.1029/2008JD009823.

*P20 L23. Explain open mode.*
**We have removed the term.**
**The term open mode is used to mention the aerosol size distribution having only the trailing side of the log-normal distribution. During the nucleation events, extremely high concentration of particles is observed in the lowest measurable size range, with 3 to 5 nm size, and concentration decreases with the increase in the size of the particle. Thus the number size distribution of freshly nucleating aerosols does not show the leading side of the distribution.**

*P21 L12. Also Hudson & Da (1996) and Hudson (2007).*
**Complied with.**

**Hudson, J.G., 2007: Variability of the relationship between particle size and cloud-nucleating ability. Geophys. Res. Let., 34, L08801, doi:10.1029/2006GL028850.**

*P21 L30-31. How can such low ks be higher than previous measurements? Fig. 2 of Hudson & Yum (2002) shows k 0.23 for 1-0.2% S.*
**Since INDOEX did not measure CCN concentration over the SEAS1 region defined in this manuscript, we can't directly compare the INDOEX values with SEAS1. However, compared to INDOEX, k values over SEAS1 are higher. SEAS1 values are closer to the continental values like at Trivandrum. Over equatorial Indian Ocean, k values measured during INDOEX is comparable to the present measurements.**

*P21 L32. They do not get activated as CCN. They are CCN.*
**We have observed that even at high supersaturations (1.0%) all CN are not activated as CCN, especially over EIO.**

*L33. This would imply that there are many CN that are not CCN, but I do not see this in this manuscript.*
**We are sorry for not giving this numbers. At 1.0% supersaturation 78±30% of CN is activated as CCN over SEAS1 and 53±28% over EIO.**

***Minor Comments:***

*Fig. 2. It would be more appropriate to reverse the positions of 0.2% and 1% in the legend.*
**Complied with.**

[Figure]

*P2. L12. Insert The before South.*
**Complied with.**

*L13. In to within.*
**Complied with.**

*L14. Delete the.*
**Complied with.**

*L23. Add Hudson & Yum (2002).*
**Complied with.**

*L28. Change never to not.*
**Complied with.**

*P3. L1-2. Add Hudson (2007).*
**Complied with.**

*L4. Delete the.*
**Complied with.**

*L5. Insert a before few. Level plural.*
**Complied with.**

*L15. Delete 2nd the.*
**Complied with.**

*P5. L32. Delete enough.*
**Complied with.**

*P6. L1. Change values to concentrations.*
**Complied with.**

*L4-5. Delete in the.*
**Complied with.**

*L9. Delete in.*
**Complied with.**

*L12. Delete 1st the.*
**Complied with.**

*L13-14. Change increased by to was. Folds singular. Change when to higher at. Move 1% before supersaturation. Replace changed from to than at. Delete to. Change increase to concentration was similar. Delete is significant.*
**Complied with.**

*P7 L1. Delete as.*
**Complied with.**

*L5. Delete 's. Add ship to relation.*
**Complied with.**

*L7. Add s to indicate.*
**Complied with.**

*L13. Insert somewhat before hydrophobic. Hydrophobic particles would not be CCN at all.*
**Complied with.**

*P8. L18. Delete s of towards.*

**Complied with.**

*P9. L12. Delete has.*
**Complied with.**

*L13. Is to was.*
**Complied with.**

*L17. Change get activated to CCN.*
**Complied with.**

*L18. Delete last the.*
**Complied with.**

*P10. L16. Insert for after accounting.*
**Complied with.**

*L18. Define coarse mode.*
**Since we don't have measurements of particles above 1µm, we have removed the term "coarse mode" from the manuscript. We have defined the fine mode (>100 nm) and ultrafine mode (<100 nm) aerosols in the manuscript. Earlier studies have shown that ultrafine aerosols have low hygroscopicity compared to the fine mode aerosols.**

*L20. Particles singular.*
**Complied with.**

*L21. Insert the after in.*
**Complied with.**

*P11. L2. Distribution plural.*
**Complied with.**

*L7. Insert the after from.*
**Complied with.**

*L12. Insert in EIO after particles. Change ed to s.*
**Complied with.**

*L13. Insert concentration after CN.*
**Complied with.**

*L23. Pattern plural.*
**Complied with.**

*P12. L7. Distribution plural.*
**Complied with.**

*L14-15. This implies relatively higher CN at EIO.*
**Though the CN concentration is several fold higher over SEAS1 compared to EIO, the activation efficiency at supersaturations higher than 0.6% is also higher over SEAS1.**

*P13. L8. Activation efficiency is redundant with CCN. It is not necessary.*
**Complied with.**

*P14. L13. Explain flip-flops.*

**This can be explained based on the critical diameter shown in Figure 10. Relatively large decrease in critical diameter with supersaturation was observed over SEAS1 compared to SEAS2 and EIO. Though the aerosol loading is high over SEAS1, CCN concentration at 0.2% supersaturation is lower over SEAS1 compared to that of EIO. High value of critical diameter (~185 nm) over SEAS1 further confirm that only a small portion of the aerosol population is activated as CCN at 0.2% supersaturation. This is mostly attributed to the presence of less hygroscopic aerosols (primary carbonaceous) above 100 nm 10 size range, which require higher supersaturation to get activated. It is interesting to notice that the critical diameter at 1.0% supersaturation is higher for EIO than SEAS1 in contrast to 0.2% and 0.4% supersaturations.**

*L13. Bursts singular.*
**Complied with.**

*L17. Delete the.*

**Complied with.**

*L23. This is so even at 0.55%.*
**We agree. we want to mention that at highest set supersaturation (1.0%), all particles are not activated over SEAS.**

*L24. It is not an increase. It is higher concentrations.*
**Complied with.**

*L26. This is not shown.*
**We estimated these values from the NSD shown in figure 7.**

*L26-7. Delete activated to. This is redundant. If they are CCN they can be activated.*
**Complied with.**

*P15. L8. Change spectra to distribution. Spectra implies CCN S spectra. Delete a. Supersaturation plural.*
**Complied with.**

*L11. Not so for 0.2 & 0.3%.*
**We agree with reviewer. The sentence is modified now. "For both high and low-UFP cases, SEAS1 aerosols are more CCN active at higher supersaturations than EIO, which is also clearly reflected in the regional mean values."**

*L 12. Move also before clearly.*
**Complied with.**

*L14. Delete system and then end the sentence.*
**Complied with.**

*L15. Delete 1st the.*
**Complied with.**

*L16. Delete parentheses and values. Insert or before k.*
**Complied with.**

*L17. Period after concentration. Change which to This. Delete in. delete last the. Delete values.*
**Complied with.**

*P16. L2. Comma after Thumba.*
**Complied with.**

*L3. Period after particles. Insert This is.*
**Complied with.**

*L5. Delete 1st the.*
**Complied with.**

*L7. Change Further to Then. Delete from the data set.*
**Complied with.**

*L8. Fig. 8a. is not mentioned.*
**Complied with.**

*L12-13. Activation fraction does not drop. It is lower.*
**Complied with.**

*P17. L11. Coefficient of determination. Delete which.*
**Complied with.**

*L12. Add ed to depend.*
**Complied with.**

*P18. L1. Change activated as to are.*
**Complied with.**

*L2. Insert at 0.4% after concentration. Change reduced drastically to lower.*
**Complied with.**

*L10-11. Specify S, probably 0.4%.*
**Complied with.**

*L12. Delete last the.*
**Complied with.**

*P19. L5. Change decreased with an increase in to was lower for higher. Supersaturation plural.*
**Complied with.**

*L6. Period after regions. Change in to This is. Change consistency to consistent. Delete the.*
**Complied with.**

*L10. Change primarily to possibly.*
**Complied with.**

*P20. L1. Insert of before aerosol. Delete highly.*
**Complied with.**

*L2. Delete 1st the. Delete size range.*
**Complied with.**

*L4. Move the after over.*

**Complied with.**

*L28. Particle plural.*
**Complied with.**

*L31. Change reinstated to emphasized.*
**Complied with.**

*L33. Change change in the aerosol to also been due to. Change as well to differences.*
**Complied with.**

*P21. L12. Insert Hudson (2007).*
**Complied with.**

*L20. Insert the before major.*
**Complied with.**

*L21. Change vice versa to not.*
**Complied with.**

*L26. Delete 1st the.*
**Complied with.**

*L30. Concentration plural. Insert k after low.*
**Complied with.**

*L32. Move at high (1.0%) supersaturations after CCN. Change get activated as to are.*
**Complied with.**

*L33. Delete even.*
**Complied with.**

*P22. L2. Change 2nd the to greater. Delete in the size distribution.*
**Complied with.**

Hoppel, W.A., Fitzgerald, J.W., & Larson, R.E. (1985). Aerosol size distributions in air masses advecting off the East Coast of the United States. Journal of Geophysical Research, 90, 2365-2379.

Hoppel, W.A., Frick, G.M., & Fitzgerald, J.W. (1996). Deducing droplet concentration and supersaturation in marine boundary layer clouds from surface aerosol measurements. Journal of Geophysical Research, 101, 26,553-26,565.

Hoppel, W.A., Frick, G.M., Fitzgerald, J.W., & Larson, R.E. (1994). Marine boundary layer measurements of new particle formation and the effects nonprecipitating clouds have on aerosol size distribution. Journal of Geophysical Research, 99, 14443–14459, doi:10.1029/94JD00797, (H94).

Hoppel, W.A., Frick, G.M. & Larson, R.E. (1986). Effect of nonprecipitating clouds on the aerosol size distribution in the marine boundary layer. Geophysical Research Letters, 13, 125-128.

Hoppel, W.A., Fitzgerald, J.W., Frick, G.M., Larson, R.E., & Mack, E.J. (1990). Aerosol size distributions and optical properties found in the marine boundary layer over the Atlantic Ocean. Journal of Geophysical Research, 95, 3659-3686.

Hudson, J.G., 2007: Variability of the relationship between particle size and cloud-nucleating ability. Geophys. Res. Let., 34, L08801, doi:10.1029/2006GL028850.

Hudson, J.G. and X. Da, 1996: Volatility and size of cloud condensation nuclei. J. Geophys. Res., 101, 4435-4442.

Hudson, J.G., Noble, S., & Tabor, S. (2015). Cloud supersaturations from CCN spectra Hoppel minima. Journal of Geophysical Research-Atmospheres, 120, 3436–3452, doi:10.1002/2014JD022669.

Hudson, J.G., S. Noble, and S. Tabor, 2018: CCN spectral shape and stratus cloud and drizzle microphysics. Journal of Geophysical Research: Atmospheres, 123, 9635-9651. http://doi.org/10.1029/2017JD027865

Noble, S.R., and Hudson, J.G., 2019: Effects of continental clouds on surface Aitken and accumulation modes. Journal of Geophysical Research: Atmospheres, 124, 5479-5502. https://doi.org/10.1029/2019JD030297.

---

## Author Comment (AC2) · 28 Jan 2020

Review of "Cloud Condensation Nuclei properties of South Asian outflow over the northern Indian Ocean during winter" by Vijayakumar S Nair et al., (ACP-2019-828)

*This paper presents some interesting results on cloud condensation nuclei (CCN) and condensation nuclei (CN) concentrations obtained over north Indian ocean as a part of ICARB-2018 conducted during winter 2018. Sophisticated data collected within the ship cruise campaign (16 Jan. 2018 to 13 Feb.2018) has been wisely used to investigate the latitudinal and longitudinal variations. Major conclusions drawn from this detailed investigations includes findings of high CCN over southeastern Arabian sea compared to equatorial Indian ocean, high CCN efficiency over south Asian outflow compared to equatorial Indian ocean, contribution of accumulation mode to high activation fraction over north Indian ocean. Further, strong association between CCN efficiency and geometrical mean diameter of aerosol number size distribution is being reported. In general, paper is concise and well written with substantial new information and apt for Atmospheric Chemistry and Physics Journal. However, few clarifications are required before accepting for its publication. Below are the some of the issues which authors need to take care. Authors are strongly encouraged to revise this manuscript.*

**We thank the reviewer for the encouraging comments and fruitful suggestions.**

Comments/Suggestions:

*Page 3, Lines 28-31, Figure 1: The spatial extend of the aerosol transport to the Indian Ocean is qualitatively depicted by the climatological (2002-2017) mean aerosol optical depth (AOD) derived from MODIS observations over the northern Indian Ocean (contours in Figure 1). These contours are hard to see from the figure. I suggest including color contour of the same.*

**Complied with. The figure is redrawn in the revised version for better clarity. Instead of multiple colorbars, we have increased the thickness of the contour and AOD values are embedded in the each contour line.**

[Figure]

*Page 4, Lines 5-7: The first phase of the cruise over southeastern Arabian Sea (SEAS) is divided into 'SEAS1' and 'SEAS2' regions where the former is influenced by the air masses from peninsular India, and the latter is from the Bay of Bengal. I wonder how the lines are drawn that differentiates SEAS1 and SEAS1.*

**We have computed the airmass back trajectories for all the days using HYSPLIT model. Three typical cases are shown in the manuscript (Figure 01). The airmass trajectories reaching at ship location above 8°N have direct influence of peninsular India while below 8°N air masses are arriving from the Bay of Bengal. Moreover, regions above 8°N has proximity to landmass and below 8°N represent the open ocean under the influence of pollution transport from Bay of Bengal.**

*Page 4, Lines 11-12: Indian sub-continent during this period due to the. This is an incomplete sentence.*

**We are sorry for the inadvertent error. The sentence is corrected in the revised version of the manuscript. "These wide spread rainfall events associated with the**

**western disturbances are also observed over the peninsular and western part of the Indian sub-continent during this period."**

*Page 17, Figure 8: I am unable to see much difference between Type 3 and Type 4. Did it qualify the followed analysis procedure?*

**As mentioned in the manuscript, we have followed stringent conditions ($R^2$>0.9) to group the similar NSDs. Though they look same in activation efficiency, there are differences in the size distributions of Type 3 and 4 especially at 10-20 nm range and 150-300 nm range. So, we have $R^2$ < 0.9 for the regression of Type 3 and Type 4. However, the difference in GMD between Type 3 and Type 4 is ~10 nm and thus activation efficiencies are comparable.**

*Page 20, Line 3: Reported 'bi-modal' size distributions should be reported 'bi-model' size distributions.*

**We have verified this in several text books (eg: Seinfeld and Pandis, Atmospheric Chemistry and Physics, 2006). Aerosol NSD with 2 modes is called as 'bimodal'.**

*There are few grammatical mistakes and typos. I suggest authors to go through the manuscript carefully again before submitting revised draft.*

**Complied with.**